# Adapting Noise to Data: Generative Flows from Learned 1D Processes

**Jannis Chemseddine** [* 1]  **Gregor Kornhardt** [* 1]  **Richard Duong** [* 1]  **Gabriele Steidl** [1]

## Abstract

The default Gaussian latent in flow-based generative models poses challenges when learning certain distributions such as heavy-tailed ones. We introduce a general framework for learning data-adaptive parametric prior distributions (latent noise) using one-dimensional quantile functions, optimized via the Wasserstein distance between noise and data. The quantile-based prior parameterization naturally adapts to both heavy-tailed and compactly supported distributions and shortens transport paths. Numerical results on heavy-tailed weather and image datasets confirm the method's flexibility and effectiveness achieved with negligible computational overhead.

## 1. Introduction

Flow-based generative models have become a dominant paradigm in modern generative modeling. In particular, score-based diffusion models (Sohl-Dickstein et al., 2015; Song & Ermon, 2019; Song et al., 2021), flow matching (FM) methods (Albergo et al., 2023; Lipman et al., 2023; Liu, 2022), and more recently few-step approaches such as consistency-style models (Song et al., 2023; Geng et al., 2025; Boffi et al., 2025a) achieve state-of-the-art performance across a wide range of domains, including image synthesis and molecular generation (Hoogeboom et al., 2022), as well as discrete modalities such as text (Austin et al., 2023).

In flow-based models, the default choice of noise distribution is Gaussian, which can cause difficulties when learning, for example, multimodal or heavy-tailed target distributions; we refer to the extensive literature (Wiese et al., 2019; Hagemann & Neumayer, 2021; Salmona et al., 2022; Pandey

[1]TU Berlin, Straße des 17. Juni 136, 10623 Berlin, Germany. Correspondence to: Jannis Chemseddine <chemseddine@math.tu-berlin.de>, Gregor Kornhardt <kornhardt@math.tu-berlin.de>, Richard Duong <duong@math.tu-berlin.de>, Gabriele Steidl <Steidl@math.tu-berlin.de>.

*Proceedings of the $43^{rd}$ International Conference on Machine Learning*, Seoul, South Korea. PMLR 306, 2026. Copyright 2026 by the author(s).

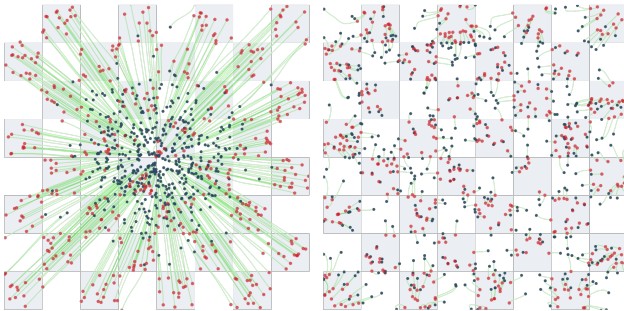

*Figure 1*. FM via optimal coupling with Gaussian noise (left) and our learned noise (right). Latent samples are shown in black, generated in red and transportation paths in green. Starting from the learned latent drastically shortens the paths.

et al., 2024; Ghane et al., 2025; Tam & Dunson, 2025). As a typical example, see the Neal's funnel in Figure 2.

To enable sampling from heavy-tailed target distributions, recent state-of-the-art approaches typically replace the standard Gaussian reference with a heavy-tailed noise distribution. However, these methods still require manually tuning the tail behavior to match the data. For instance, (Pandey et al., 2024) proposes a Student-$t$ noise distribution with a tunable degrees-of-freedom parameter $\nu$. Similarly, (Shariatian et al., 2025a) extends the SDE framework to heavy-tailed settings by driving the dynamics with $\alpha$-stable noise.

In this paper, we propose a novel *lightweight* approach to design a noise distribution tailored to the data by *learning* it directly from samples. Motivated by the componentwise independent structure of the Gaussian and the Brownian motion, we construct *multidimensional* flows from *one-dimensional* ones. This enables us to address the latter ones by their *quantile functions*. While quantile functions can represent any 1D probability distribution, their monotonicity makes them particularly simple to parameterize, for example by using rational quadratic splines (Gregory & Delbourgo, 1982). More precisely, for quantile functions $Q^i$, $i = 1, \ldots, d$, we consider the *quantile processes*

$$X_t^i = (1 - t) X_0^i + t \, Q^i(U^i), \qquad t \in [0, 1],$$

with i.i.d. $U^1, \ldots, U^d \sim \mathcal{U}[0, 1]$. We learn individual quantile functions $Q_\phi^i$ such that their componentwise concatenation $\mathbf{Q}_\phi(\mathbf{U}) \coloneqq (Q_\phi^1(U^1), ..., Q_\phi^d(U^d))$ is "*close*" to the data by minimizing the Wasserstein distance between the data distribution and the noise. Note that this is a constrained

optimization problem, since the noise is learned as a product of 1D measures. The simplicity of quantile functions gives us a flexible tool, which enables us to simultaneously learn the noising process and apply the FM framework.

Our approach allows us to i) avoid the limitations of Gaussian base noise, ii) eliminate manual fine-tuning of the noise distribution, and iii) drastically shorten the resulting transport paths as illustrated in Figure 1. Further, it can be extended to fit into few-step models, e.g. inductive moment matching (IMM) (Zhou et al., 2025; Boffi et al., 2025b) using more general mean reverting processes.

**Contributions.** 1. We introduce a general construction method for generative neural flows by decomposing multi-dimensional flows into one-dimensional components and applying a general mean reverting process. Ultimately, this allows us to work with 1D noising processes in the FM framework.

2. We examine interesting 1D noising processes besides Brownian motion: the physics-inspired Kac process and the MMD gradient flow, leading to compactly supported noise and a better regularity of the FM velocity field.

3. We show how our framework can be generalized for few-step models via so-called quantile interpolants. Furthermore, we offer an alternative method to construct multi-dimensional processes from one-dimensional ones using radially symmetric processes.

4. As main contribution, we propose to *learn the 1D noise distributions* within the FM framework in a data adapted way, by parameterizing them through quantile functions and employing the Wasserstein distance.

5. Numerical experiments demonstrate that our method efficiently handles diverse marginal structures including heavy-tailed, compact, and multi-modal distributions. Learned quantiles shorten transport paths by capturing per-coordinate structure, while delegating cross-dimensional dependencies to the velocity field. For heavy-tailed datasets, the quantile prior removes the need to hand-tune the noise parameters while yielding consistently better results.

## 2. Background on Flow Matching

In general, flow models can be described mathematically via curves in Wasserstein space, see, e.g. (Ambrosio et al., 2008) and (Lipman et al., 2024; Wald & Steidl, 2025). In the following, we denote the target distribution by $\mu_0 \in \mathcal{P}(\mathbb{R}^d)$ and the noise distribution by $\mu_1 \in \mathcal{P}(\mathbb{R}^d)$.

---

[1]Note that we used the independent coupling for training of these models. We also used z-score normalization.

### 2.1. Curves in Wasserstein Spaces

Let $(\mathcal{P}_2(\mathbb{R}^d), W_2)$ denote the complete metric space of probability measures with finite second moments equipped with the Wasserstein distance

$$W_2^2(\mu, \nu) := \min_{\pi \in \Pi(\mu,\nu)} \int_{\mathbb{R}^d \times \mathbb{R}^d} \|x - y\|^2 \, \mathrm{d}\pi(x, y)$$

Here $\Pi(\mu, \nu)$ denotes the set of all probability measures on $\mathbb{R}^d \times \mathbb{R}^d$ having marginals $\mu$ and $\nu$. By $\pi_o \in \Pi(\mu, \nu)$ we denote the minimizer of the right-hand side. The push-forward measure of $\mu \in \mathcal{P}_2(\mathbb{R}^d)$ by a measurable map $\mathcal{T} : \mathbb{R}^d \to \mathbb{R}^d$ is defined by $\mathcal{T}_\sharp \mu := \mu \circ \mathcal{T}^{-1}$. A narrowly continuous curve $\mu_t : [0, 1] \to \mathcal{P}_2(\mathbb{R}^d)$ is absolutely continuous, iff there exists a Borel measurable vector field $v : [0, 1] \times \mathbb{R}^d \to \mathbb{R}^d$ with $\|v_t\|_{L_2(\mathbb{R}^d, \mathbb{R}^d, \mu_t)} \in L_2([0, 1])$ such that $(\mu_t, v_t)$ satisfies the continuity equation

$$\partial_t \mu_t + \nabla_x \cdot (\mu_t v_t) = 0 \tag{1}$$

in the sense of distributions.

### 2.2. From the Continuity Equation to the ODE

If the vector field fulfills $\int_0^1 \sup_{x \in B} \|v_t(x)\| + \mathrm{Lip}(v_t, B) \, \mathrm{d}t < \infty$ for all compact $B \subset \mathbb{R}^d$, then the ODE

$$\partial_t \varphi(t, x) = v_t(\varphi(t, x)), \qquad \varphi(0, x) = x, \tag{2}$$

has a solution $\varphi : I \times \mathbb{R}^d \to \mathbb{R}^d$ and $\mu_t = \varphi(t, \cdot)_\sharp \mu_0$.

Starting in the target distribution $\mu_0$ and ending in a simple latent distribution $\mu_1$, as usual in diffusion models, we can reverse the flow from the latent to the target distribution in inference using just the opposite velocity field $-v_{1-t}$ in the ODE (2). If the velocity field $v_t$ is learned, we can sample from the target distribution by starting in a sample from the latent one and then applying our favorite ODE solver in (2).

### 2.3. Flow Matching: Learning the Velocity Field

Flow matching learns the velocity field $v_t$ with a neural network $v_t^\theta$ by minimizing the loss

$$\mathcal{L}(\theta) := \mathbb{E}_{t, x \sim \mu_t} \left[ \left\| v_t^\theta(x) - v_t(x) \right\|^2 \right]$$

with uniformly sampled time in $[0, 1]$. Indeed, this intractable loss can be rewritten as $\mathcal{L}(\theta) = \mathcal{L}_{\mathrm{CFM}}(\theta) + \text{const}$ with the conditional flow matching (CFM) loss

$$\mathcal{L}_{\mathrm{CFM}}(\theta) := \mathbb{E}_{t, x_0 \sim \mu_0, x \sim \mu_t(\cdot | x_0)} \left[ \left\| v_t^\theta(x) - v_t(x | x_0) \right\|^2 \right].$$

*Remark* 2.1 (Flow Matching with Couplings.). Consider a coupling $\pi \in \Pi(\mu_0, \mu_1)$ and the induced curve $\mu_t := (e_t)_\sharp \pi$ with $e_t(x, y) := (1 - t)x + ty$. Then it is known that $\mathcal{L}_{\mathrm{CFM}}(\theta)$ can be rewritten as

$$\mathcal{L}_{\mathrm{CFM}}(\theta) = \mathbb{E}_{t, (x,y) \sim \pi} \left[ \left\| v_t^\theta \big((1 - t)x + ty\big) - (y - x) \right\|_2^2 \right].$$

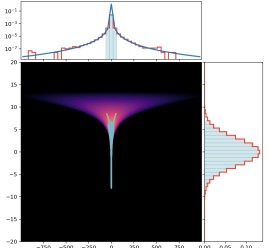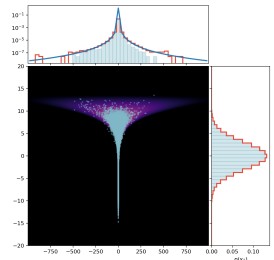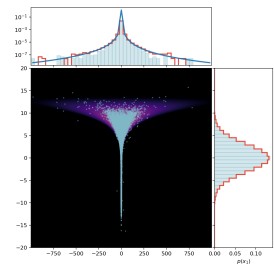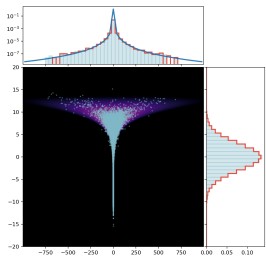

*Figure 2.* Sampling of Neal's funnel with different latent distributions.[1] Left to Right: *uniform* on $[-1, 1]$, *standard Gaussian, Student-T* (with parameters $(20, 4)$ inspired by the choice in (Pandey et al., 2024)) and our *learned distribution*. The last two heavy-tailed noises perform significantly better.

Taking the optimal coupling $\pi_o$ instead of the usually applied independent coupling $\pi = \mu_0 \otimes \mu_1$, leads to a reduced variance in training and both shorter and straighter paths, see (Tong et al., 2024; Pooladian et al., 2023).

### 2.4. Velocity Fields via Stochastic Processes

Absolutely continuous curves and their velocity fields can be alternatively derived via stochastic processes: for a probability space $(\Omega, \Sigma, \mathbb{P})$ and a differentiable stochastic processes $(\mathbf{X}_t)_{t \in [0,1]}$ with $\mathbf{X}_t \in L_2(\Omega, \mathbb{R}^d, \mathbb{P})$, we have that $(\mu_t, v_t)$ with

$$\mathbf{X}_t \sim \mu_t := \mathbf{X}_{t,\sharp}\mathbb{P} \quad \text{and} \quad v_t := \mathbb{E}[\dot{\mathbf{X}}_t | \mathbf{X}_t = \cdot] \qquad (3)$$

satisfies the continuity equation. We introduce a special mean reverting process which generalizes previous standard concepts, see Remark 2.2.

**Mean Reverting Processes.** Let us consider a continuously differentiable (noising) process $(\mathbf{N}_t)_t$ with $\mathbf{N}_0 \equiv 0 \in \mathbb{R}^d$ and $\mathbf{N}_t \sim \mu_t^{\mathbf{N}}$ with associated velocity field $v_t^{\mathbf{N}}$. In the next section, we will show how to get access to such velocity fields. Then we define the *mean-reverting* process by

$$\mathbf{X}_t := f(t)\,\mathbf{X}_0 + \mathbf{N}_{g(t)}, \quad t \in [0,1], \qquad (4)$$

with smooth *scheduling functions* $f, g$ fulfilling

$$f(0) = 1, \quad f(1) = 0 \quad \text{and} \quad g(0) = 0, \quad g(1) = 1. \quad (5)$$

Then $\mathbf{X}_1 = \mathbf{N}_1$ and the process $\mathbf{X}_t$ starts in $\mathbf{X}_0$. Differentiation of (4) results in

$$\dot{\mathbf{X}}_t = \dot{f}(t)\,\mathbf{X}_0 + \dot{g}(t)\,\dot{\mathbf{N}}_{g(t)}.$$

Then, the conditional velocity field of $\mathbf{X}_t$ is given by

$$\begin{aligned}
v_t^{\mathbf{X}}(x \mid x_0) &= \mathbb{E}\big[\dot{\mathbf{X}}_t \mid \mathbf{X}_t = x, \ \mathbf{X}_0 = x_0\big] \\
&= \mathbb{E}\big[\dot{f}(t)\,x_0 + \dot{g}(t)\,\dot{\mathbf{N}}_{g(t)} \mid \mathbf{N}_{g(t)} = x - f(t)x_0\big] \\
&= \dot{f}(t)\,x_0 + \dot{g}(t)\,v_{g(t)}^{\mathbf{N}}\big(x - f(t)x_0\big). \qquad (6)
\end{aligned}$$

Now, the conditional flow matching loss $\mathcal{L}_{\mathrm{CFM}}$ can be minimized regarding $\mathbf{X}_t \sim \mu_t$: namely sampling $y \sim \mathbf{N}_{g(t)}$ we obtain a sample $x \sim (\mathbf{X}_t \mid \mathbf{X}_0 = x_0)$ from (4) satisfying

$$v_t^{\mathbf{X}}(x \mid x_0) = \dot{f}(t)\,x_0 + \dot{g}(t)\,v_{g(t)}^{\mathbf{N}}(y).$$

*Remark* 2.2 (Relation to FM and Diffusion). Consider the stochastic process

$$\mathbf{X}_t^{\mathrm{FM}} = \alpha_t \mathbf{X}_0 + \sigma_t \mathbf{X}_1, \qquad \mathbf{X}_1 \sim \mathcal{N}(0, I_d). \quad (7)$$

Choosing $f(t) := \alpha_t$, $g(t) := \sigma_t^2$ and the standard Brownian motion $\mathbf{N}_t = \mathbf{W}_t$, it holds the equality in distribution

$$\mathbf{X}_t^{\mathrm{FM}} \overset{d}{=} f(t)\mathbf{X}_0 + \mathbf{W}_{g(t)} = \mathbf{X}_t.$$

Then $f(t) := 1 - t$, $g(t) := t^2$ yields (independent) FM (Lipman et al., 2023), and $f(t) := \exp\left(-\frac{h(t)}{2}\right)$, $g(t) := 1 - \exp\left(-h(t)\right)$, where $h(t) := \int_0^t \beta_{\min} + s(\beta_{\max} - \beta_{\min})\,\mathrm{d}s$ with, e.g., $\beta_{\min} = 0.1$, $\beta_{\max} = 20$, corresponds to processes used in score-based generative models (Song et al., 2021), see Appendix B.

## 3. Reduction to One-Dimensional Flows

Recall that in standard FM and diffusion models, isotropic Gaussian noise is gradually added to the data, i.e.

$$\mathbf{X}_t = \alpha_t \mathbf{X}_0 + \mathbf{N}_t, \qquad (8)$$

where $\mathbf{N}_t = (N_t^1, \ldots, N_t^d)$ with i.i.d. $N_t^i \sim \mathcal{N}(0, \sigma_t^2)$. Motivated by this, we restrict ourselves to noising processes $\mathbf{N}_t$ that decompose into one-dimensional components. This allows us to propose a general construction method for accessible *conditional* flows in FM.

Let $N_t^1, \ldots, N_t^d$ be a family of independent 1D stochastic processes with laws $\mu_t^i \in \mathcal{P}_2(\mathbb{R})$. For each $i = 1, \ldots, d$, let $v_t^i \colon \mathbb{R} \to \mathbb{R}$ be the associated velocity field such that the pair $(\mu_t^i, v_t^i)$ satisfies the one-dimensional continuity

equation (1). Then, the law of the $d$-dimensional process $\mathbf{N}_t := (N_t^1, \ldots, N_t^d)$ is the product measure

$$\mu_t(x) \;=\; \prod_{i=1}^{d} \mu_t^i(x^i), \quad x = (x^1, \ldots, x^d) \in \mathbb{R}^d, \quad (9)$$

and the corresponding $d$-dimensional velocity field is given componentwise, see (Duong et al., 2025).

**Proposition 3.1.** *Let $\mu_t$ be given by* (9)*, where the $\mu_t^i$ are absolutely continuous curves in $\mathbb{R}$ with velocity fields $v_t^i$. Then $\mu_t$ satisfies a multi-dimensional continuity equation* (1) *with a velocity field which decomposes into the univariate velocities $v_t(x) := \big(v_t^1(x^1), \ldots, v_t^d(x^d)\big)$.*

In summary, assuming access to the 1D velocities, we can **construct accessible conditional flows for FM** as follows:

1.  *One-dimensional noise:* Start with a 1D process and an associated curve $\mu_t$ with $\mu_0 = \delta_0$, $0 \in \mathbb{R}$ and an accessible velocity field $v_t$ in the 1D continuity equation

    $$\partial_t \mu_t + \partial_x(\mu_t v_t) = 0, \qquad \mu_0 = \delta_0. \quad (10)$$

2.  *Multi-dimensional noise:* Set up a multi-dimensional conditional flow model starting in $\mu_0 = \delta_0$, $0 \in \mathbb{R}^d$ with possibly different, but independent 1D processes as described in (9).

3.  *Incorporating the data:* Construct a multi-dimensional conditional flow model starting in $\mu_0 = \delta_{x_0}$ for any data point $x_0 \sim \mu_0$ by mean-reversion as shown in Section 2.4.

*Remark* 3.2 (Learning Cross Dimensional Correlation of the Data.)*.* As usual in diffusion or flow matching models, the velocity field (3) (or score function in diffusion) is responsible for learning cross-dimensional correlations and dependencies of the target $\mathbf{X}_0$.

Concerning step 1, we explore three interesting 1D (noising) processes in connection with their respective PDEs in the Appendix A, namely:

- Wiener process $W_t$ and diffusion equation,
- Kac process $K_t$ and damped wave equation,
- Uniform process $U_t$ and MMD gradient flow.

In each case, we explicitly calculate the respective conditional measure flow and its conditional velocity field in Appendix A, such that the conditional flow matching loss $\mathcal{L}_{\mathrm{CFM}}$ can be minimized. In each case, the absolutely continuous curve starting in $\delta_0$ and the corresponding velocity field can be calculated analytically. In contrast to the usual Wiener process $W_t$ in diffusion and FM models, the latter two processes $K_t, U_t$ do not enjoy a trivial analogue in

multiple dimensions: in case of $K_t$ the corresponding PDE (damped wave equation) is no longer mass-conserving in dimension $d \geq 3$, see (Tautz & Lerche, 2016); in case of $U_t$ the mere existence of the MMD gradient flow in multiple dimensions is unclear due to the lack of convexity of the MMD, see (Hertrich et al., 2024). Our general construction method makes these 1D processes accessible for generative modeling in arbitrary dimensions, hinting at a wide range of suitable noising processes. For example, the recent work (Han et al., 2025) utilizes the Kac framework for model distillation.

As an alternative to above componentwise construction, we describe an approach using radially symmetric processes in Appendix D.

## 4. One-Dimensional Flows with Quantiles

We have outlined how to construct accessible conditional flows for FM using 1D noising processes. So far, we have considered three fixed 1D processes. However, every 1D distribution can be described by its quantile functions. This will enable us in Section 5 to construct *arbitrary noising processes* and to learn the noise, see also (Chemseddine et al., 2026).

To this end, we revisit the connection between 1D distributions and their quantile functions. Furthermore, we introduce quantile processes and quantile interpolants.

### 4.1. Background on Quantile Functions

The restriction to componentwise noising processes $\mathbf{N}_t = (N_t^1, \ldots, N_t^d)$ in (8)[2] allows us to use the quantile functions of the 1D components. The *cumulative distribution function* (CDF) $R_\mu$ of $\mu \in \mathcal{P}_2(\mathbb{R})$ and its *quantile function* $Q_\mu$ are given by

$$R_\mu(x) := \mu\big((-\infty, x]\big), \quad x \in \mathbb{R}, \quad (11)$$
$$Q_\mu(u) := \min\{x \in \mathbb{R} : R_\mu(x) \geq u\}, \quad u \in (0, 1).$$

The quantile functions form a closed, convex cone

$$\mathcal{C} := \{f \in L_2(0, 1) : f \text{ increasing } a.e.\} \quad (12)$$

in $L_2(0, 1)$. The mapping $\mu \mapsto Q_\mu$ is an isometric embedding of $(\mathcal{P}_2(\mathbb{R}), W_2)$ into $(L_2(0, 1), \|\cdot\|_{L_2})$, meaning that

$$W_2^2(\mu, \nu) = \int_0^1 \big|Q_\mu(s) - Q_\nu(s)\big|^2 \, \mathrm{d}s$$

and $\mu = Q_{\mu,\sharp}\mathcal{U}_{(0,1)}$. Let $U \sim \mathcal{U}[0, 1]$ be uniformly distributed on $[0, 1]$. Now, any probability measure flow $\mu_t$ can be described by their quantile flow $Q_t := Q_{\mu_t}$, such

---

[2]Besides componentwise 1D processes we may also use triangular decompositions, not addressed in this paper.

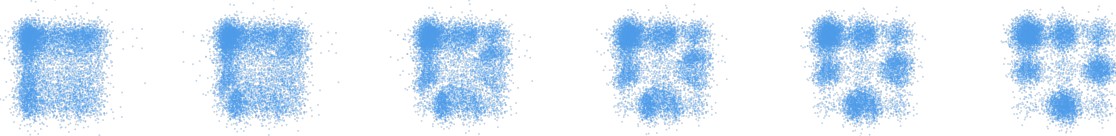

*Figure 3.* A generated trajectory from the learned quantile latent (left) to the unevenly weighted Gaussian mixture target (right). The *adapted latent* $\mathbf{Q}_\phi$ (left image) is already close to the target distribution $\mathbf{X}_0 \sim \mu_0$.

that $\mu_t = Q_{t,\sharp}\mathcal{U}_{(0,1)}$ and $Q_t \circ U$ is a stochastic process with marginals $\mu_t$.

**Parameterization.** Quantile functions are both universal by being able to express any 1D distribution, and simple to parametrize, since they only need to satisfy the requirements from (12). For example quantile functions can be modeled using rational quadratic splines (Gregory & Delbourgo, 1982; Durkan et al., 2019), as described in Section 6.1. Moreover, Dirac measures $\delta_a$ are easily represented in this framework via constant quantile functions $Q \equiv a$.

### 4.2. Quantile Processes

We can therefore model any *multi-dimensional* noising process, that decomposes into its components $\mathbf{N}_t = (N_t^1, \ldots, N_t^d)$, via quantile functions. Namely let $X_0$ be any component $\mathbf{X}_0^i$ of $\mathbf{X}_0 \sim \mu_0$, and $f, g : [0,1] \to \mathbb{R}$ smooth schedules fulfilling (5). We assume that we are given a flow $(Q_t)_t$ of quantile functions $Q_t : (0,1) \to \mathbb{R}$, $t \in [0,1]$, which fulfill $Q_0 \equiv 0$. Let $U \sim \mathcal{U}(0,1)$. We introduce the *quantile process*

$$Z_t = f(t)X_0 + Q_{g(t)}(U), \quad t \in [0,1]. \tag{13}$$

By choosing the quantile process $Q_t^i(U) \overset{d}{=} t \cdot Q_{\mathcal{N}(0,1)}(U)$, the resulting process coincides with the standard Gaussian interpolation process in (8).

### 4.3. Quantile Interpolants.

Assuming $Q_t$ is invertible on its image, the inverse is given by the CDF $R_t : Q_t(0,1) \to [0,1]$. Let us briefly mention how our setting fits into the framework of consistency models. To this end, we define the *quantile interpolants* for $s, t \in [0,1]$ by

$$I_{s,t}(x, y) = f(s)x + Q_{g(s)}\big(R_{g(t)}(y - f(t)x)\big). \tag{14}$$

This generalizes the interpolants used in Denoising Diffusion Implicit Models (DDIM), see Remark C.1.

**Proposition 4.1.** *For all $x, y \in \mathbb{R}$ and all $s, r, t \in [0,1]$, it holds $I_{0,t}(x, y) = x$, $I_{t,t}(x, y) = y$, and*

$$I_{s,r}(x, I_{r,t}(x, y)) = I_{s,t}(x, y).$$

*Furthermore, inserting the quantile process (13) yields $I_{s,t}(Z_0, Z_t) = Z_s$.*

The proof is given in Appendix C. Proposition 4.1 allows us to also apply the concept of consistency models to our quantile process (13). In the Appendix C, we demonstrate this by means of the recently proposed *inductive moment matching* (IMM) (Zhou et al., 2025).

## 5. Adapting Noise to Data

As noted in many papers, see also the related work in Section 7, the choice of the noise distribution can have a significant impact on sampling performance. In Figure 1, we visualize the path length under the learned vector field for the checkerboard distribution. In Figure 2, we evaluate the Neal's Funnel (Neal, 2003) distribution under different noise distributions, ranging from compact-support to heavy-tailed.

Rather than manually selecting a noise family and tuning its parameters, we propose to *learn* the noise (prior distribution) directly from the data. We focus on the simple setting $\mathbf{Q}_{g(t)} = g(t)\mathbf{Q}$ and schedules $f(t) = 1 - t$, $g(t) = t$. This corresponds to the standard linear interpolation in FM with conditional velocity $v_t^{\mathbf{Q}}(x) = x/t$; see Appendix A.4.

We restrict to noise $\mathbf{Q}$ with independent components having law from the set

$$S := \Big\{\nu \in \mathcal{P}_2(\mathbb{R}^d) : \nu(x) = \prod_{i=1}^d \nu^i(x^i)\Big\},$$

corresponding to quantile processes of the form

$$X_t^i = (1 - t) X_0^i + t Q^i(U^i), \quad t \in [0,1],$$

with $\mathbf{Q}(\mathbf{u}) := (Q^1(u^1), \ldots, Q^d(u^d))$. The quantile functions $Q^i$ determine the scale, support, and tail behavior of each marginal of $\mathbf{Q}(\mathbf{U})$.

**Learning Objective.** We learn the noise $\mathbf{Q}_\phi$ by minimizing the Wasserstein distance between $\mu_0$ and $\nu_\phi := (\mathbf{Q}_\phi)_{\#}\mathcal{U}([0,1]^d)$,

$$\mathcal{L}_{\text{AN}}(\phi) = W_2^2(\mu_0, \nu_\phi). \tag{15}$$

Note that due to the restriction of our quantiles to the class $S$, the minimizer of (15) is in general *not* $\mu_0$. Crucially, the independence constraint restricts $(\mathbf{Q}_\phi)_{\#}\,\mathcal{U}([0,1]^d)$ to

per-coordinate adaptation and prevents encoding *cross-dimensional* correlations. The latter are introduced via the optimal transport coupling $(x, y)$ and modeled by the velocity field through the target $(y - x)$. By this separation the latent remains simple and computationally efficient, while delegating dependencies to the flow.

**Entropy Regularization.** In high-dimensional settings and given fixed batch sizes, the signal for the quantile function can be noisy, potentially leading to degenerate solutions. To mitigate this, we add a regularization term to the loss that penalizes the expected negative log-determinant of the Jacobian of the quantile. Since the quantile maps from a uniform distribution $U \sim \mathcal{U}[0, 1]$, this term equals the differential entropy $h$ of the learned latent

$$\mathcal{R}(\phi) := h(\mathbf{Q}_\phi(U)) = h(U) + \mathbb{E}[\log|\det J_{\mathbf{Q}_\phi}(U)|]$$
$$= \mathbb{E}[\log|\det J_{\mathbf{Q}_\phi}(U)|].$$

Access to analytic derivatives makes this efficient, for more details see E.2.

**Joint Training Objective.** Although our quantiles can be trained independently, in order to provide an aligned training signal for the velocity field, we propose to train $\mathbf{Q}_\phi$ *jointly* with velocity $v^\theta$. Hence, we aim to minimize the loss

$$\mathcal{L}(\theta, \phi) = \mathcal{L}_{\text{CFM}}(\theta, \phi) + \lambda \mathcal{L}_{\text{AN}}(\phi) - \beta \mathcal{R}(\phi), \quad (16)$$

where $\beta \in [0, \infty)$ is a hyperparameter controlling the regularizer, for stability of this choice see Figure 6 and Section G, and $\lambda$ is weighting the quantile loss. The velocity field training loss $\mathcal{L}_{\text{CFM}}(\theta, \phi)$ is given by

$$\mathbb{E}_{t,(x,y_\phi)\sim\pi_\phi}\left[\left\|v_t^\theta\big((1-t)x + ty_\phi\big) - (\text{sg}(y_\phi) - x)\right\|_2^2\right],$$

where $\pi_\phi \in \Pi_o(\mu_0, \nu_\phi)$ and $\text{sg}(\cdot)$ denotes the stop-gradient operator. We visualize the effect of choosing $\lambda = 0$ in Figure 18. The stop-gradient on $X_1 - X_0$ means that the quantile network only receives gradients through the interpolated states $X_t = (1-t)X_0 + tX_1$. This discourages trivial collapse, since the quantile cannot reduce the FM loss simply by shrinking the endpoint displacement.

**Implementation Details.** In practice, we optimize via minibatches; see Appendix F.2 for details. We compute a minibatch OT map $T$ minimizing $\sum_{j=1}^B \|\mathbf{x}_0^{(j)} - \mathbf{y}^{(T(j))}\|_2^2$ for batches from $\mathbf{X}_0$ and $\mathbf{Q}_\phi(\mathbf{U})$. Crucially, this coupling is reused both for optimizing the quantile and for OT-FM, see Remark 2.1. We train the quantile jointly for a fixed number of iterations before freezing its parameters; thereafter only the velocity field is optimized.

---

**Algorithm 1** Joint learning of 1D quantiles and FM velocity

**Input:** dataset $\mathcal{D}$, batch size $B$, weights $\lambda, \beta$, iterations $K$; quantile model $\mathbf{Q}_\phi$, velocity model $v^\theta$
**for** $k = 1$ **to** $K$ **do**
  Sample $\{\mathbf{x}_i\}_{i=1}^B \sim \mathcal{D}$, $\{\mathbf{u}_j\}_{j=1}^B \sim \mathcal{U}([0,1]^d)$, $\{t_j\}_{j=1}^B \sim \mathcal{U}(0,1)$
  $C_{ij} \leftarrow \|\mathbf{x}_i - \mathbf{Q}_\phi(\mathbf{u}_j)\|_2^2$
  $T \leftarrow \arg\min_T \sum_{i=1}^B C_{i,T(i)}$
  Define $P$ such that $P(j) = i$ iff $T(i) = j$
  **for** $j = 1$ **to** $B$ **do**
    $\mathbf{z}_j \leftarrow (1 - t_j)\mathbf{x}_{P(j)} + t_j\,\mathbf{Q}_\phi(\mathbf{u}_j)$
    $\mathbf{v}_{\text{target},j} \leftarrow \text{sg}\big(\mathbf{Q}_\phi(\mathbf{u}_j) - \mathbf{x}_{P(j)}\big)$
  **end for**
  $\widehat{\mathcal{L}}_{\text{AN}} \leftarrow \frac{1}{B}\sum_{j=1}^B \|\mathbf{x}_{P(j)} - \mathbf{Q}_\phi(\mathbf{u}_j)\|_2^2$
  $\widehat{\mathcal{R}} \leftarrow \frac{1}{B}\sum_{j=1}^B \log\left|\det J_{\mathbf{Q}_\phi}(\mathbf{u}_j)\right|$
  $\widehat{\mathcal{L}}_{\text{CFM}} \leftarrow \frac{1}{B}\sum_{j=1}^B \|v^\theta(\mathbf{z}_j, t_j) - \mathbf{v}_{\text{target},j}\|_2^2$
  $\widehat{\mathcal{L}} \leftarrow \widehat{\mathcal{L}}_{\text{CFM}} + \lambda\,\widehat{\mathcal{L}}_{\text{AN}} - \beta\,\widehat{\mathcal{R}}$
  Update $(\theta, \phi)$ by a gradient step on $\widehat{\mathcal{L}}$
**end for**
**Output:** learned parameters $(\theta, \phi)$

---

# 6. Experiments

To provide intuition and validate our proposed method, we conduct experiments on synthetic, image and weather datasets. First we briefly outline how to efficiently parametrize quantile functions. The code is available at https://github.com/TUB-Angewandte-Mathematik/Adapting-Noise.

### 6.1. Rational Quadratic Splines

We parameterize each quantile function $Q^i$ using a rational quadratic spline (RQS) (Gregory & Delbourgo, 1982; Durkan et al., 2019). Unlike normalizing flows that compose many RQS layers within coupling architectures, we use a single RQS per coordinate with directly optimized parameters.

For coordinate $i$, the spline $S_\phi^i : [-B, B] \to [-B, B]$ is defined by $K$ knots with associated bin widths $\{w_k\}_{k=1}^K$, bin heights $\{h_k\}_{k=1}^K$, and slopes $\{s_k\}_{k=0}^K$. We enforce positivity via softplus and normalize widths and heights to span $2B$. This guarantees strict monotonicity. Outside $[-B, B]$, we extend $S_\phi^i$ linearly with $C^1$ continuity at the boundaries.

To map from $(0, 1)$ to the spline domain, we apply $\psi(u) = \text{logit}(u)$ or $\psi(u) = B(2u - 1)$, yielding the quantile

$$Q_\phi^i(u) = a_i \cdot S_\phi^i(\psi(u)) + b_i,$$

where the scale $a_i > 0$ and bias $b_i$ are learned per coordinate.

**Computational Overhead.** The total parameter count is $\mathcal{O}(Kd)$: for $K = 32$ bins and $d = 3072$ (CIFAR-10), approximately 300k parameters. This lightweight parametrization, combined with reusing the minibatch OT coupling and freezing the quantile after 55k iterations, introduces minimal overhead compared to standard Gaussian OT-FM. On CIFAR-10 (NVIDIA RTX 5090), we measure approximately $2.7\%$ overhead during joint training and $0.5\%$ after freezing.

## 6.2. Synthetic Datasets

We begin by qualitatively analyzing our algorithm on several synthetic 2D distributions, see also Appendix F.1, each designed to highlight a specific aspect of our approach. We provide intuition about the learned latent distribution and demonstrate that it is closer to the data in the Wasserstein sense, yields shorter transport paths, and successfully captures the tail behavior.

**Gaussian Mixture Model (GMM).** We first consider a 2D GMM with nine unevenly weighted modes, as visualized in Figure 3. Due to the independence assumption inherent in our factorized quantile function, the learned latent cannot perfectly replicate the target's joint distribution and is *not the product of the correct marginals*, see also Example E.1. Instead, it approximates a distribution where the components cannot further independently improve the transport cost to the target.

**Funnel Distribution.** The funnel distribution (Figure 2) presents a challenge due to its heavy-tailed, conditional structure. Several methods have been proposed in the diffusion context (Pandey et al., 2024; Shariatian et al., 2025a;b). We compare to (Pandey et al., 2024), where Student-$t$ parameters were hand-selected per dimension. Using a capacity-constrained network (three layers, width 64, no positional embeddings), we observe that a compact latent performs worst, followed by the Gaussian, while our learned latent successfully *adapts to the target's heavy tails* (see Figure 12).[3]

**Checkerboard Distribution.** The checkerboard in Figure 11 features compact support. Our method learns a latent approximating a uniform distribution over the target's support. Combined with OT coupling, the resulting *transport paths are substantially shorter* than starting from a Gaussian as visualized in Figure 1. In addition, in training, the vector field converges faster, see Figure 13. This underscores our central claim: combining a data-dependent latent with a data-dependent coupling can significantly improve

---

[3]Due to the high variance when sampling minibatch from the funnel, we pre-train the quantile and use the independent coupling for all models.

performance.

## 6.3. Image Datasets

Next, we analyze our method on standard image generation benchmarks. Our quantile is extremely lightweight compared to the U-Net architecture used for the flow model. In Table 2, the statistical fit between the noise and data distributions is compared for the Gaussian and learned quantile noise.

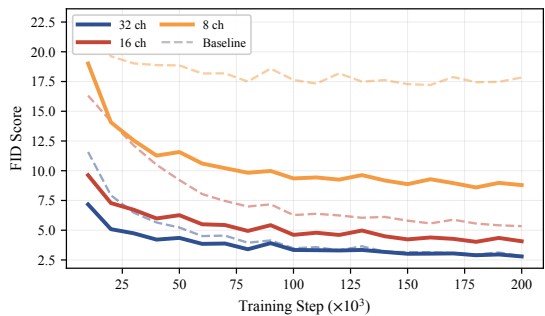

*Figure 4.* Ablation on U-Net capacity for MNIST using channels $8, 16, 32$. The FID curves show that our method achieves significantly lower FIDs when using less channels.

**MNIST.** MNIST exhibits strong marginal structure: center pixels are frequently active while border pixels are nearly always zero. Our learned quantile captures these statistics, concentrating mass in regions corresponding to active pixels as shown in Figure 5. Where a pixel is essentially always black, the learned quantile concentrates around that value; in center regions with higher uncertainty, the quantiles remain spread around zero (gray), accurately reflecting the data variability. We compare learned and empirical quantiles at different pixel locations in Figure 15.

This dataset is well-suited for our method since the marginal structure provides a strong learning signal. In Figure 4, we compare performance under capacity constraints using channels $8, 16, 32$ ($\lambda = 1$, $\beta = 0.1$). Our learned latent achieves significantly lower FIDs across all capacity levels. See Figure 16 in the Appendix for the total amount of parameters used for each model. By removing redundant information in the noise distribution, the network can allocate its parameters more efficiently toward modeling the relevant structure. The independence assumption prevents capturing spatial correlations (e.g., digit shapes).

**CIFAR-10.** We evaluate on CIFAR-10 to demonstrate that our method scales to natural images and integrates seamlessly with standard architectures. We follow the setup of (Tong et al., 2024), using the U-Net architecture from (Nichol & Dhariwal, 2021) with $35M$ parameter, to enable a fair comparison.

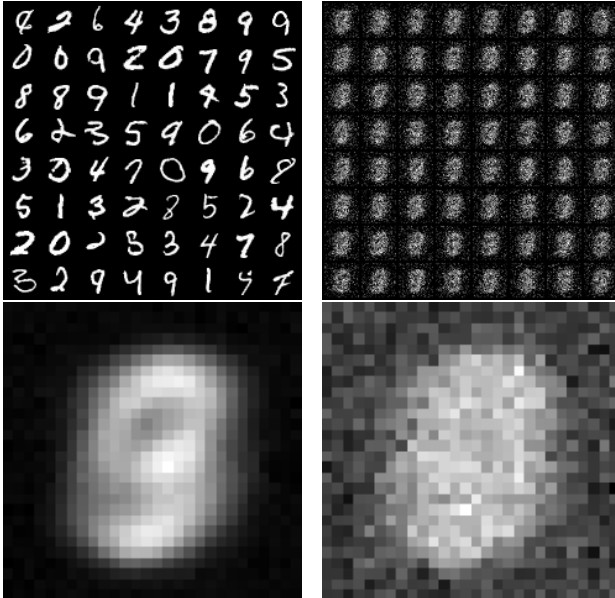

*Figure 5.* Left to Right: Generated samples, samples from the learned latent and mean and standard deviation of the learned latent.

Figure 6 reports results for varying $\beta$ at fixed $\lambda = 1$. Our method successfully learns a data-adapted latent while maintaining competitive FID scores, confirming that the quantile training remains stable at scale. See also Figure 18 for details on the training stability. Improvements are marginal as expected, since CIFAR-10 features strong spatial and inter-channel correlations that a product measure cannot fully capture. Using a larger 55M-parameter model, we improve the FID to **3.25** for the quantile prior, compared to 3.37 for the Gaussian one.

### 6.4. Weather Dataset

**HRRR-mini.** We include a large scale dataset with strongly heavy-tailed precipitation statistics, we are using the *HRRR-mini* (Mardani et al., 2024; NVIDIA Corporation) dataset, which consists of 100k samples, where each pixel corresponds to a spatial resolution of around 3km. The resolution is $64 \times 64$, and consists of temperature, east-west and north-south wind, and total precipitation per pixel. The main objective is to correctly model the tail behavior of the total precipitation.

For this experiment, we use exactly the same setup and U-Net network with 35M parameters, and only interchange the noise distribution. The experiment configurations are in F.6. The models are compared on their performance in matching the tail statistics of the true precipitation data. For this, the following metrics are used; for an extended description, see Section F.6. The *extreme event frequency error* is the relative error for $\mathbb{P}(X \geq q_{0.999})$, where $q_{0.999}$

is the value of the quantile at the 99.9% percentile. The *extreme event magnitude error* is the relative error in the conditional expectation $\mathbb{E}[X \mid X \geq q_{0.999}]$. The *spectral distance* is the $\ell_1$ distance between the radially averaged log power spectra. The last three metrics are also used in (Pandey et al., 2024) and are defined as follows. The *tail KS distance* is the mean KS statistic restricted to beyond the 0.1% and 99.9% percentiles. The *kurtosis deviation* is given by $|1 - k_{\text{gen}}/k_{\text{real}}|$, and the *skewness deviation* is given by $|1 - \gamma_{\text{gen}}/\gamma_{\text{real}}|$.

| Metric | Gaussian baseline | Student-$t$ baseline | Quantile (Ours) |
|---|---|---|---|
| Extreme event frequency error ↓ | 0.9689 | 0.8859 | **0.7550** |
| Extreme event magnitude error ↓ | 0.2455 | 0.1482 | **0.0634** |
| Spectral distance ↓ | 3.1836 | 2.0719 | **1.1063** |
| Tail KS distance ↓ | 0.2067 | 0.1014 | **0.0393** |
| Kurtosis deviation ↓ | 4.930 | 2.890 | **1.588** |
| Skewness deviation ↓ | 1.157 | 0.830 | **0.580** |

*Table 1.* Tail-centric HRRR64-Mini precipitation evaluation. Lower is better for all metrics.

In Table 1, the different noise distributions are compared. As already discussed theoretically in other papers (see Section 7 on related work), using a noise distribution with light tails does not work well for heavy-tailed data. In our experiment, we obtain the same result: the Gaussian noise produces much worse results than the heavy-tailed Student's $t$ and the learned noise distribution. Compared with the Student's $t$, our learned quantile distribution learns a heavy-tailed distribution without the need to hand-tune it along every dimension, as is required for the Student's $t$. The learned noise clearly outperforms the other two noise distributions.

## 7. Related Work

**Generative Models.** Current state-of-the-art methods largely rely on corrupting images with Gaussian noise. Diffusion models (Song et al., 2021; Ho et al., 2020; Song et al., 2022) and the closely related flow matching models (Liu et al., 2023; Lipman et al., 2023; Albergo et al., 2023) form the basis of most approaches. To further improve inference speed, few-step methods have achieved remarkable results (Song et al., 2023; Geng et al., 2025; Zhou et al., 2025).

**Learning the Flow.** The following papers keep the Gaussian distribution as a starting point, but updated the interpolation path. There exist only few approaches to learn the noising process, (Bartosh et al., 2025) fit the forward diffusion process to the backward via a learned invertible map that is trained end-to-end, (Kapusniak et al., 2024) use metric flow matching, i.e., a neural network to adapt the path to a underlying Riemannian metric. In a related approach (Sa-

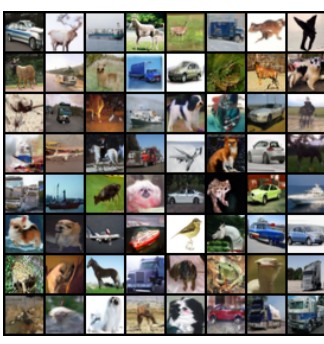
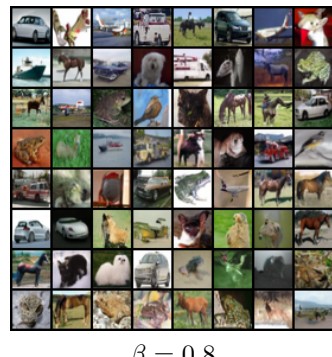

| Euler steps → | **20** | **100** |
|---|---|---|
| $\beta$-logdet ↓ | **FID** | **FID** |
| 0.2 | 7.81 | 4.75 |
| 0.3 | **7.48** | 4.53 |
| 0.5 | 7.66 | 4.49 |
| 0.8 | 7.77 | **4.42** |
| 1.0 | 8.35 | 4.66 |
| **Baseline** | 8.42 | 4.63 |

Baseline    $\beta = 0.8$

*Figure 6.* CIFAR results for a selection of regularization parameters and for the baseline, for complete results see Figure 17. Our method reached the best validation FID after 320k steps, while the baseline took 340k. We evaluated the FID using 5 seeds and report the mean. We used those checkpoints for the evaluation. The visualized samples were generated using 100 Euler steps.

hoo et al., 2024) learns a input-conditioned componentwise Gaussian noise schedule. In the setting of sampling from unnormalized target densities, (Blessing et al., 2025a) learn the latent noise by optimizing the mean and covariance of a Gaussian prior. A Gaussian mixture prior has been used both for sampling (Blessing et al., 2025b) and for generative modeling (Issachar et al., 2025).

Along the lines of normalizing flows and VAEs, there is a large body of work on learning more flexible noise distributions and priors (Stimper et al., 2022; Bauer & Mnih, 2019; Hickling & Prangle, 2025; Amiri et al., 2024).

**Limitations of Gaussian Noise.** From a complementary perspective, (Wiese et al., 2019) propose to separate marginal modeling from dependence structure using copula and marginal flows, recognizing that standard architectures struggle with tail asymptotics, a motivation conceptually aligned with our componentwise quantile approach. On the other hand (Pandey et al., 2024; Zhang et al., 2024) design heavy-tailed diffusions using Student-$t$ latent distributions, and (Shariatian et al., 2025b) extend the framework to the family of $\alpha$-stable distributions.

Recent work by (Ghane et al., 2025) explains why heavy-tailed sampling can be challenging: diffusion models driven by Gaussian noise satisfy a concentration-of-measure property. In particular, when initialized from a Gaussian distribution, the generated distribution inherits many Gaussian-like properties. Moreover, by (Tam & Dunson, 2025), GANs, VAEs and diffusion models with Gaussian or log-concave noise distributions can only generate light-tailed samples and are not universal generators. See Figure 2 for a heavy-tailed example, where Gaussian noise fails.

## 8. Conclusions

We provide a "quantile sandbox" for building generative models: a unifying theory and a practical toolkit that turns noise selection into a data-driven design element. Our con-

struction plugs seamlessly into standard objectives including flow matching and few-step models. Furthermore, our experiments demonstrate that it is possible to learn extremely efficient, freely parametrized latent distributions beyond the usual smooth transformations of Gaussians. Our work opens promising directions for future research. Extensions include developing well suited objectives to learn time-dependent quantile functions to optimize the entire path distribution, or designing more sophisticated conditional quantile functions for tasks like class-conditional or text-to-image generation.

## Acknowledgments

GS and RD acknowledge funding by the German Research Foundation (DFG) within the Excellence Cluster MATH+ and JC by project STE 571/17-2 within the The Mathematics of Deep Learning. GK acknowledges funding by the BMBF VIScreenPRO (ID: 100715327).

## Impact Statement

This paper presents work whose goal is to advance the field of machine learning. There are many potential societal consequences of our work, none of which we feel must be specifically highlighted here.

## Limitations

Our learned latent is a product distribution and therefore does not model correlated noise across dimensions. While our experiments include CIFAR-10 and HRRR64, we have not systematically tested scaling to very high-dimensional settings.

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

# A. Examples of One-Dimensional Flows

We provide three interesting examples, namely the well-established diffusion flow, the recently proposed Kac flow, and the Wasserstein gradient flow of the MMD functional with negative absolute distance kernel towards a uniform measure. Paths of the processes are depicted in Figure 7 and their probability flows are shown in Figures 8, 9 and 10.

In each case, the absolutely continuous curve $\mu_t$ starting in $\delta_0$ (e.g. conditional) and the corresponding velocity field can be given analytically. Note that in the latter two cases, multi-dimensional generalizations of the flows are not trivially given, which further underlines the strength of our 1D approach. Henceforth, if the measures $\mu_t$ admit a density function, we will denote it by $p_t$.

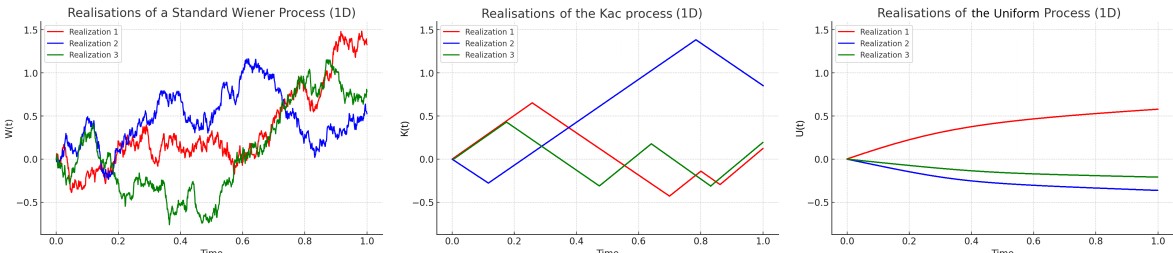

*Figure 7.* Three realisations of a standard Wiener process (left), the Kac process (middle), and the Uniform process (right), simulated until time $T = 1$.

## A.1. Wiener Process and Diffusion Equation

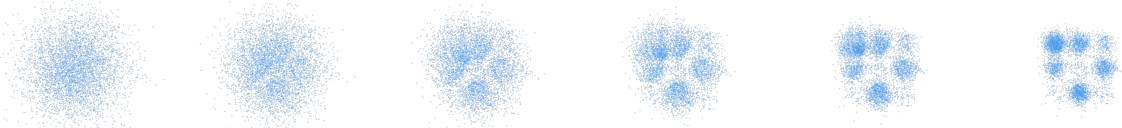

*Figure 8.* A generated trajectory from a Flow Matching model trained using the conditional density and velocity given by the Wiener process. As described in Section 2.4 we define the mean reverting process and use schedules $f(t) = 1 - t$ and $g(t) = t^2$.

First, consider the standard Wiener process (Brownian motion) $(W_t)_t$ starting in 0 whose probability density flow $p_t$ is given by the solution of the diffusion equation

$$\partial_t p_t = \nabla \cdot (p_t \frac{1}{2} \nabla \log p_t) = \frac{1}{2} \Delta p_t, \quad t \in (0, 1], \qquad \lim_{t \downarrow 0} p_t = \delta_0, \tag{17}$$

where the limit for $t \downarrow 0$ is taken in the sense of distributions. The solution is analytically known to be

$$p_t(x) = (2\pi t)^{-\frac{d}{2}} e^{-\frac{\|x\|^2}{2t}}.$$

Thus, the latent distribution is just the Gaussian $p_1 = \mathcal{N}(0, I_d)$. The velocity field in (17) reads as

$$v_t(x) = -\frac{1}{2} \nabla \log p_t = \frac{x}{2t}. \tag{18}$$

However, its $L_2$-norm fulfills $\|v_t\|_{L_2(\mathbb{R}, p_t)}^2 = \frac{d}{4t}$, and is therefore not integrable over time, i.e. $\|v_t\|_{L_2(\mathbb{R}, p_t)} \notin L_2(0, 1)$. In practice, instability issues caused by this explosion at times close to the target need to be avoided by e.g. time truncations, see e.g. (Kim et al., 2022). For a heuristic analysis also including drift-diffusion flows, we refer to (Pidstrigach, 2022). Note that in the case of diffusion, there is no significant distinction between the uni- and multivariate setting.

## A.2. Kac Process and Damped Wave Equation

The Kac process (Kac, 1974), also known as persistent random walk, originates from a discrete random walk, which starts in 0 and moves with velocity parameter $c > 0$ in one direction until it reverses its direction with probability $a \Delta_t$, $a > 0$. A

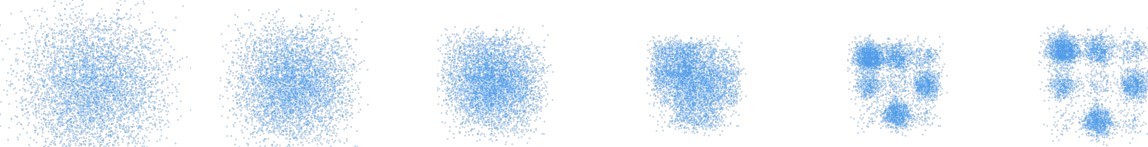

*Figure 9.* A generated trajectory from a flow matching model trained using the conditional density and velocity given by the Kac process with $(a, c) = (9, 3)$. As described in Section 2.4 we define the mean reverting process and use schedules $f(t) = 1 - t$ and $g(t) = t^2$.

continuous-time analogue is given by the Kac process which is defined using the homogeneous *Poisson point process* $N_t$ with rate $a$, i.e. i) $N_0 = 0$; ii) the increments of $N_t$ are independent, iii) $N_t - N_s \sim \text{Poi}\big(a(t - s)\big)$ for all $0 \leq s < t$. Now the *Kac process starting in* $0$ is given by

$$K_t := \text{B}_{\frac{1}{2}} \, c \int_0^t (-1)^{N_s} \, \mathrm{d}s, \tag{19}$$

where $\text{B}_{\frac{1}{2}} \sim \text{Ber}(\frac{1}{2})$ is a Bernoulli random variable[4] taking the values $\pm 1$. Note that in contrast to diffusion processes, the Kac process $K_t$ *persistently* maintains its linear motion between changes of directions (jumps of $N_t$), see Figure 7.

By the following proposition, the Kac process is related to the damped wave equation, also known as telegrapher's equation, and its probability distribution admits a computable vector field such that the continuity equation is fulfilled. For a proof we refer to (Duong et al., 2025).

**Proposition A.1.** *The probability distribution flow of* $(K_t)_t$ *admits a singular and absolutely continuous part via*

$$\mu_t(x) = \frac{1}{2} e^{-at} \big( \delta_0(x + ct) + \delta_0(x - ct) \big) + \tilde{p}_t(x), \tag{20}$$

*with the absolutely continuous part*

$$\tilde{p}_t(x) := \frac{1}{2} e^{-at} \Big( \beta ct \frac{I_0'(\beta r_t(x))}{r_t(x)} + \beta I_0(\beta r_t(x)) \Big) 1_{[-ct, ct]}(x), \qquad r_t(x) := \sqrt{c^2 t^2 - x^2},$$

*where* $\beta := \frac{a}{c}$, *and* $I_0$ *denotes the* 0-*th* modified Bessel function of first kind. *The distribution* (20) *is the generalized solution of the damped wave equation*

$$\partial_{tt} u(t, x) + 2a \, \partial_t u(t, x) = c^2 \partial_{xx} u(t, x), \tag{21}$$
$$u(0, x) = \delta_0(x), \qquad \partial_t u(0, x) = 0.$$

*Further* $(\mu_t, v_t)$ *solves the continuity equation* (10) *where the velocity field is analytically given by*

$$v_t(x) := \begin{cases} \dfrac{x}{t + \frac{r_t(x)}{c} \frac{I_0(\beta r_t(x))}{I_0'(\beta r_t(x))}} & \text{if} \quad x \in (-ct, ct), \\ c & \text{if} \quad x = ct, \\ -c & \text{if} \quad x = -ct, \\ \text{arbitrary} & \text{otherwise.} \end{cases}$$

*The Kac velocity field admits the boundedness* $\|v_t\|_{L_2(\mathbb{R}, \mu_t)} \leq c$, *and hence,* $\|v_t\|_{L_2(\mathbb{R}, \mu_t)} \in L_2(0, 1)$.

Interestingly, the damped wave equation (21) is closely related to the diffusion equation via Kac' insertion method. It is based on the following theorem, whose proof based on semigroup theory can be found in (Griego & Hersh, 1971), see also (Janssen, 1990; Kac, 1974).

**Theorem A.2.** *For any initial function* $f_0 \in H^2(\mathbb{R}^d)$, $d \geq 1$, *let* $w_c(t, x)$ *be the solution of the* undamped *wave equation with velocity* $c > 0$ *given by*

$$\partial_{tt} w(t, x) = c^2 \Delta w(t, x), \quad x \in \mathbb{R}^d, \, t > 0,$$
$$w(0, x) = f_0(x), \qquad \partial_t w(0, x) = 0.$$

---

[4]More precisely, $\text{B}_{\frac{1}{2}}$ is *two-point* distributed with values $\{-1, 1\}$.

*Then, the functions defined by*

$$h(t,x) := \mathbb{E}\left[w_1\left(\sigma W_t, x\right)\right], \quad resp. \quad u(t,x) := \mathbb{E}\left[w_c(c^{-1}S_t, x)\right]$$

*solve the diffusion equation*

$$\partial_t h(t,x) = \frac{\sigma^2}{2}\Delta h(t,x), \quad x \in \mathbb{R}^d, \ t > 0,$$
$$h(0,x) = f_0(x),$$

*resp. the multi-dimensional damped wave equation*

$$\partial_{tt} u(t,x) + 2a\,\partial_t u(t,x) = c^2\Delta u(t,x), \quad x \in \mathbb{R}^d, \ t > 0,$$
$$u(0,x) = f_0(x), \qquad \partial_t u(0,x) = 0. \tag{22}$$

As a consequence, it is not hard to show the following corollary, see (Duong et al., 2025).

**Corollary A.3.** *For any $t \geq 0$, the solution $u^{a,c}(t,\cdot)$ of the damped wave equation* (22) *converges to the solution $h(t,\cdot)$ of the diffusion equation for $a, c \to \infty$ with fixed $\sigma^2 = \frac{c^2}{a}$.*

In other words, diffusion can be seen as *"an infinitely $a$-damped wave with infinite propagation speed $c$"*. Note that the diffusion-related concept of particles traveling with infinite speed violates Einstein's laws of relativity and has therefore found resistance in the physics community (Cattaneo, 1958; Chester, 1963; Vernotte, 1958; Tautz & Lerche, 2016).

We also like to stress that in multiple dimensions, the damped wave equation (21) is *no longer* mass-conserving as in 1D (Tautz & Lerche, 2016), and hence eludes a characterization via stochastic processes. Figure 9 shows the generation of samples from a weighted Gaussian Mixture Model (GMM) using Flow Matching and the Kac process as our noising process. As described in Section 2.4 we define the mean reverting process and use schedules $f(t) = 1 - t$ and $g(t) = t^2$.

### A.3. Uniform Process and MMD Gradient Flow

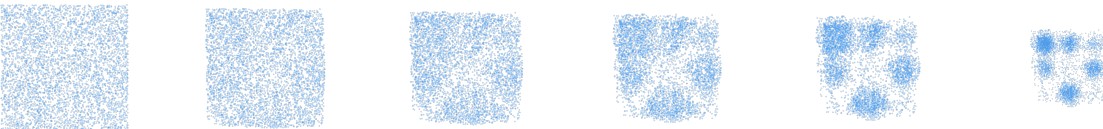

*Figure 10.* A generated trajectory from a flow matching model trained using the conditional density and velocity given by the MMD gradient flow. As described in Section 2.4 we define the mean reverting process and use schedules $f(t) = 1 - t$ and $g(t) = t$.

Wasserstein gradient flows are special absolutely continuous measure flows whose velocity fields are negative Wasserstein (sub-)gradients of functionals $\mathcal{F}_\nu$ on $\mathcal{P}_2(\mathbb{R}^d)$ with the unique minimizer $\nu$. The gradient descent flow should reach this minimizer as $t \to \infty$. In this context, the MMD functional with the non-smooth negative distance kernel $K(x,y) = -|x-y|$ given by

$$\mathcal{F}_\nu(\mu) = \mathrm{MMD}_K^2(\mu,\nu) := -\frac{1}{2}\int_{\mathbb{R}^2} |x-y|\,\mathrm{d}\left(\mu(x) - \nu(x)\right)\mathrm{d}\left(\mu(y) - \nu(y)\right), \tag{23}$$

stands out for its flexible flow behavior between distributions of different support (Hertrich et al., 2024). In 1D, its Wasserstein gradient flow $\mu_t$ can be equivalently described by the flow of its quantile functions $Q_{\mu_t}$ with respect to an associated functional on $L_2(0,1)$. Note that the MMD functional (23) loses its convexity (along generalized geodesics) in multiple dimensions (Hertrich et al., 2024), and the general existence of their Wasserstein gradient flows is unclear in the multivariate case. This yields another reason to work in 1D, where we have have the following proposition.

**Proposition A.4.** *The Wasserstein gradient flow $\mu_t$ of the MMD functional* (23) *starting in $\mu_0 = \delta_0$ towards the uniform distribution $\nu = \mathcal{U}[-b,b]$ with fixed $b > 0$ reads as*

$$\mu_t = \left(1 - \exp(-\tfrac{t}{b})\right)\mathcal{U}[-b,b],, \qquad t > 0, \tag{24}$$

*with corresponding velocity field*

$$v_t(x) = \frac{x}{b\left(\exp\left(\frac{t}{b}\right) - 1\right)}, \quad x \in \mathrm{supp}(\mu_t). \tag{25}$$

*It holds $\|v_t\|^2_{L_2(\mathbb{R},\mu_t)} = \frac{2b}{3}\exp(-\frac{2t}{b})$, and hence, $\|v_t\|_{L_2(\mathbb{R},\mu_t)} \in L_2(0,1)$. A corresponding (stochastic) process $(U_t)_t$ is given by $U_t := b\left(1 - \exp\left(-\frac{t}{b}\right)\right)U$, where $U \sim \mathcal{U}[-1,1]$, such that $\mathrm{Law}(U_t) = \mu_t$.*

We prove the proposition more general for $\nu = \mathcal{U}[a,b]$ and a flow starting in $x_0 \in [a,b]$, i.e. we show

$$\mu_t = \mathcal{U}\left[a + (x_0 - a)\exp\left(-r(t)\right), b - (b - x_0)\exp\left(-r(t)\right)\right], \quad t > 0 \tag{26}$$

with $r(t) := \frac{2t}{b-a}$ and

$$v_t(x) = \frac{2}{b-a}\left(\frac{x - x_0}{\exp(r(t)) - 1}\right). \tag{27}$$

To this end, we need the relation between measures in $\mathcal{P}_2(\mathbb{R})$ and cumulative distribution functions, see (11). For $\nu = \mathcal{U}[a,b]$, we have that

$$R_\nu(x) = \begin{cases} 0, & \text{if } x < a, \\ \frac{x-a}{b-a}, & \text{if } a \le x \le b, \\ 1, & \text{if } x > b \end{cases}$$

and $Q_\nu(s) = a(1-s) + bs$. In (Hertrich et al., 2024) it was shown that the functional $F_\nu\colon L_2(0,1) \to \mathbb{R}$ defined by

$$F_\nu(u) := \int_0^1\left((1-2s)\big(u(s) + Q_\nu(s)\big) + \int_0^1 |u(s) - Q_\nu(t)|\,\mathrm{d}t\right)\mathrm{d}s \tag{28}$$

fulfills $\mathcal{F}_\nu(\mu) = F_\nu(Q_\mu)$ for all $\mu \in \mathcal{P}_2(\mathbb{R})$. Moreover, we have the following equivalent characterization of Wasserstein gradient flows of $\mathcal{F}_\nu$, which can be found in Theorem 4.5 (Duong et al., 2024).

**Theorem A.5.** *Let $\mathcal{F}_\nu$ and $F_\nu$ be defined by (23) and (28), respectively. Then the Cauchy problem*

$$\begin{cases} \partial_t g(t) \in -\partial F_\nu(g(t)), & t \in (0,\infty), \\ g(0) = Q_{\mu_0}, \end{cases}$$

*has a unique strong solution $g$, and the associated curve $\gamma_t := (g(t))_\#\Lambda_{(0,1)}$ is the unique Wasserstein gradient flow of $\mathcal{F}_\nu$ with $\gamma(0+) = (Q_{\mu_0})_\#\Lambda_{(0,1)}$. More precisely, there exists a velocity field $v_t^*$ such that $(\gamma_t, v_t^*)$ satisfies the continuity equation (10), and it holds the relations*

$$v_t^* \circ g(t) \in -\partial F_\nu(g(t)) \quad \text{and} \quad v_t^* \in -\partial\,\mathcal{F}_\nu(\gamma_t). \tag{29}$$

Lastly note that here, the subdifferential $\partial F_\nu(u)$ is explicitly given by the singleton

$$-\partial F_\nu(u) = -\nabla F_\nu(u) = 2(\cdot - R_\nu \circ u) \quad \text{for all } u \in L_2(0,1),$$

see Lemma 4.3 (Duong et al., 2024).

*Proof of Proposition A.4.* We want to apply Theorem A.5 to $(\mu_t, v_t)$ in (26) and (27). The uniform distribution in (26) has the quantile function

$$Q_{\mu_t}(s) = \left(1 - \exp\left(-r(t)\right)\right)\left(a + (b-a)s\right) + x_0\exp\left(-r(t)\right), \quad s \in (0,1).$$

For all $t > 0$ and all $s \in (0,1)$, we have $Q_{\mu_t}(s) \in [a,b]$ since $x_0 \in [a,b]$, and thus

$$
\begin{aligned}
-\nabla F_\nu(Q_{\mu_t})(s) &= 2s - 2r_\nu(Q_{\mu_t}(s)) \\
&= 2s - 2\frac{\left(1 - \exp\left(-r(t)\right)\right)\left(a + (b-a)s\right) + x_0\exp\left(-r(t)\right) - a}{b-a} \\
&= 2\left(s - \frac{x_0 - a}{b-a}\right)\exp\left(-r(t)\right).
\end{aligned}
$$

On the other hand, it holds

$$\partial_t Q_{\mu_t}(s) = -2\frac{x_0 - a}{b - a}\exp\left(-r(t)\right) - \frac{(-2)(b - a)s}{b - a}\exp\left(-r(t)\right) = 2\left(s - \frac{x_0 - a}{b - a}\right)\exp\left(-r(t)\right).$$

By Theorem A.5, $(\mu_t)$ is the unique Wasserstein gradient flow of $\mathcal{F}_\nu$ starting in $\delta_0$.

Furthermore, there exists a velocity field $v_t^*$ satisfying the continuity equation (10) and the relations (29). For $s \in (0, 1)$ and $t > 0$, let $y := g_s(t) = a + (x_0 - a)\exp\left(-r(t)\right) + (b - a)\left(1 - \exp\left(-r(t)\right)\right)s$. Then, we have $s = \frac{y - a - (x_0 - a)\exp(-r(t))}{(b - a)(1 - \exp(-r(t)))}$, and thus by (29),

$$
\begin{aligned}
v_t^*(y) = v_t^*(Q_{\mu_t}(s)) &= 2\left(s - \frac{x_0 - a}{b - a}\right)\exp\left(-r(t)\right) \\
&= 2\left(\frac{y - a - (x_0 - a)\exp\left(-r(t)\right)}{(b - a)\left(1 - \exp\left(-r(t)\right)\right)} - \frac{x_0 - a}{b - a}\right)\exp\left(-r(t)\right) \\
&= \frac{2}{b - a}\left(\frac{y - a - (x_0 - a)}{1 - \exp\left(-r(t)\right)}\right)\exp\left(-r(t)\right) \\
&= \frac{2}{b - a}\left(\frac{y - x_0}{\exp\left(r(t)\right) - 1}\right)
\end{aligned}
$$

for all $y \in g_s(t)(0, 1) = [a + (x_0 - a)\exp(-r(t)), b - (b - x_0)\exp\left(-r(t)\right)]$. Lastly, let us compute the action. For $t > 0$ we have

$$
\begin{aligned}
\|v_t\|_{L^2(\mathbb{R}, \mu_t)}^2 &= \int_{a + (x_0 - a)\exp\left(-\frac{2t}{b-a}\right)}^{b - (b - x_0)\exp\left(-\frac{2t}{b-a}\right)} \frac{4(x - x_0)^2}{(b - a)^2\left(\exp\left(\frac{2t}{b-a}\right) - 1\right)^2} \frac{1}{(b - a)\left(1 - \exp\left(-\frac{2t}{b-a}\right)\right)}\,\mathrm{d}x \\
&= \frac{4}{(b - a)^3\left(\exp\left(\frac{2t}{b-a}\right) - 1\right)^2\left(1 - \exp\left(-\frac{2t}{b-a}\right)\right)} \int_{a + (x_0 - a)\exp\left(-\frac{2t}{b-a}\right)}^{b - (b - x_0)\exp\left(-\frac{2t}{b-a}\right)} (x - x_0)^2\,\mathrm{d}x \\
&= \frac{4}{(b - a)^2\exp\left(-\frac{2t}{b-a}\right)\left(\exp\left(\frac{2t}{b-a}\right) - 1\right)^3}\left[\frac{(x - x_0)^3}{3}\right]_{a + (x_0 - a)\exp\left(-\frac{2t}{b-a}\right)}^{b - (b - x_0)\exp\left(-\frac{2t}{b-a}\right)} \\
&= \frac{4\left(1 - \exp\left(-\frac{2t}{b-a}\right)\right)^3}{3(b - a)^2\exp\left(-\frac{2t}{b-a}\right)\left(\exp\left(\frac{2t}{b-a}\right) - 1\right)^3}\left[(b - x_0)^3 - (a - x_0)^3\right] \\
&= \frac{4\left[(b - x_0)^3 - (a - x_0)^3\right]}{3(b - a)^2}\exp\left(-\frac{4t}{b - a}\right).
\end{aligned}
$$

and the proof is finished. $\qquad\square$

Note that the fact that $v_t^*$ is uniquely determined on $\operatorname{supp}\mu_t = \overline{g_t(0, 1)}$, correlates with the fact that the gradient $v_t^* \circ g(t) = -\nabla F_\nu(g(t))$ is a *singleton*. Outside of $\operatorname{supp}\mu_t$, the velocity field may be arbitrarily extended, which yields a velocity $\tilde{v}_t \in -\partial \mathcal{F}_\nu(\mu_t)$ in a *non-singleton* subdifferential. The velocity $v_t^*$ may be *uniquely* chosen from the tangent space $T_{\mu_t}\mathcal{P}_2(\mathbb{R})$, or equivalently, by choosing it to have minimal norm, i.e. $v_t^* \equiv 0$ outside of $\operatorname{supp}\mu_t$.

### A.4. Scaled Latent Distributions

Finally, we consider a simple class of processes obtained by a deterministic scaling of a latent random variable. In particular, we will see that the above Wiener process and the Uniform process are of this form, while the Kac process is not. Let $Z$ be a random variable with law $\rho_Z \in \mathcal{P}_2(\mathbb{R})$, and let $g\colon [0, 1] \to [0, \infty)$ be continuously differentiable with $g(0) = 0$ and $g(1) = 1$. We consider

$$Y_t := g(t)\,Z, \qquad t \in [0, 1],$$

with $Y_t \sim \mu_t$. Supposing that $\mu_t$ has density $\rho_t$, we get

$$\rho_t(x) = g(t)^{-d} \rho_Z\left(\frac{x}{g(t)}\right), \qquad t > 0, \quad \text{and} \quad \lim_{t \downarrow 0} \mu_t = \delta_0.$$

Then straightforward computation yields that $\mu_t$ together with the velocity field

$$v_t(x) = \frac{g'(t)}{g(t)} x, \qquad x \in \text{supp}(\mu_t)$$

with the convention $v_t(0) = 0$ and arbitrary outside $\text{supp}(\mu_t)$, solves the continuity equation (10). Further, it holds

$$\int_0^1 \|v_t\|_{L_2(\mathbb{R},\mu_t)}^2 \, \mathrm{d}t = \mathbb{E}[\|Z\|^2] \int_0^1 \left(g'(t)\right)^2 \mathrm{d}t < \infty \quad \text{whenever } g' \in L_2(0,1).$$

Also note that if $Z = Q(U)$ for a quantile function $Q : (0,1) \to \mathbb{R}$ and a random variable $U \sim \mathcal{U}([0,1])$, we have

$$\mathbb{E}[\|Z\|^2] = \int_0^1 |Q(u)|^2 \, \mathrm{d}u,$$

i.e. the second moment of $Z$ is exactly given by the $L_2$-norm of its quantile. Hence, explosions of the velocity's norm $\int_0^1 \|v_t\|_{L_2(\mathbb{R},\mu_t)}^2 \, \mathrm{d}t$ can be directly controlled by the derivative of the time schedule $g$, and the size of the quantile function $Q$ of the latent variable.

The Wiener process fits into this framework with $g(t) = \sqrt{t}$ and $Z \sim \mathcal{N}(0,1)$, which recovers the exploding vector field $v_t(x) = \frac{1}{2t}x$ in (18). Also the Uniform process appears as a special case of the scaling process. In contrast, the Kac process does *not* belong to this class, as it is not generated by a deterministic scaling map but by persistent velocity switching, cf. (19).

## B. Flow Matching as Special Mean Reverting Processes

### B.1. The Gaussian Case

Let us shortly verify that our componentwise approach using the mean-reverting process (4), i.e.

$$\mathbf{X}_t := f(t)\,\mathbf{X_0} + \mathbf{Y}_{g(t)},$$

leads to the usual flow matching objective. where we choose the scheduling functions $f(t) := 1 - t$, $g(t) := t^2$, the target random variable $\mathbf{X}_0 \sim \mu_0$, and a standard Wiener process $\mathbf{Y}_t$ in $\mathbb{R}^d$ (independent of $\mathbf{X}_0$): First, it holds $\mathbf{Y}_{t^2} \sim \mathcal{N}(0, t^2 I_d)$, hence $\mathbf{Y}_{t^2} \overset{d}{=} t\,\mathbf{Z}$ with $\mathbf{Z} \sim \mathcal{N}(0, I_d)$, so that

$$\mathbf{X}_t \overset{d}{=} (1-t)\mathbf{X_0} + t\,\mathbf{Z}.$$

Furthermore, by (18) the 1D components of $\mathbf{Y}_t$ admit the velocity field $v_t^i(x^i) = \frac{x^i}{2t}$, $x^i \in \mathbb{R}$, and by Proposition 3.1 the multi-dimensional process $\mathbf{Y}_t$ admits the velocity field $v_{\mathbf{Y}}(t, x) = (\frac{x^1}{2t}, ..., \frac{x^d}{2t}) = \frac{x}{2t}$, $x = (x^1, ..., x^d) \in \mathbb{R}^d$. By the calculation (6), the conditional velocity field corresponding to $\mathbf{X}_t$ starting in $x_0 \in \mathbb{R}^d$ reads as

$$\begin{aligned}
v_{\mathbf{X}}(t, x \mid x_0) &= \dot{f}(t)\, x_0 + \dot{g}(t)\, v_{\mathbf{Y}}\big(g(t), x - f(t)x_0 \mid 0\big) \\
&= -x_0 + 2t\, v_{\mathbf{Y}}\big(t^2, x - (1-t)x_0 \mid 0\big) \\
&= -x_0 + \frac{x - (1-t)x_0}{t}.
\end{aligned}$$

Now, if $x \sim P_{\mathbf{X}_t}(\cdot \mid x_0)$, i.e. $x = (1-t)x_0 + tz$ with $z \sim \mathcal{N}(0, I_d)$, then it follows

$$v_{\mathbf{X}}(t, x \mid x_0) = -x_0 + \frac{(1-t)x_0 + tz - (1-t)x_0}{t} = z - x_0, \tag{30}$$

which is the usual constant-in-time conditional FM velocity along the straight-line trajectories between $x_0 \sim \mu_0$ and $z \sim \mathcal{N}(0, I_d)$.

## B.2. The Uniform Case

Now consider any component of the mean-reverting process (4) with $f(t), g(t)$ to be chosen, $X_0$ being a component of $\mathbf{X}_0 \sim \mu_0$, and $Y_t$ given by the MMD gradient flow (24), i.e. $Y_t := b\left(1 - \exp\left(-\frac{t}{b}\right)\right) U$, where $U \sim \mathcal{U}[-1, 1]$. Let $v_Y$ be the corresponding velocity field from (25). Then, we have

$$
\begin{aligned}
v_X(t, x|x_0) &= \dot{f}(t)\, x_0 \;+\; \dot{g}(t)\, v_Y\big(g(t),\, |x - f(t)x_0|\big) \frac{x - f(t)x_0}{|x - f(t)x_0|} \\
&= \dot{f}(t)\, x_0 \;+\; \dot{g}(t)\, \frac{x - f(t)x_0}{b\left(\exp\left(\frac{g(t)}{b}\right) - 1\right)}.
\end{aligned}
$$

Now, along the trajectory $x \sim P_{X_t}(\cdot \mid x_0)$, i.e.

$$
x \;=\; f(t) x_0 + b\left(1 - \exp\left(-\frac{g(t)}{b}\right)\right) u \;=:\; \alpha_t\, x_0 + \sigma_t\, u, \tag{31}
$$

with $u \sim \mathcal{U}(-1, 1)$, the velocity calculates as

$$
\begin{aligned}
v_X(t, x \mid x_0) &= \dot{f}(t)\, x_0 \;+\; \dot{g}(t)\, \frac{b\left(1 - \exp\left(-\frac{g(t)}{b}\right)\right) u}{b\left(\exp\left(\frac{g(t)}{b}\right) - 1\right)} \\
&= \dot{f}(t)\, x_0 \;+\; \dot{g}(t) \exp\left(\frac{-g(t)}{b}\right) u \\
&= \dot{\alpha}_t\, x_0 + \dot{\sigma}_t\, u, \tag{32}
\end{aligned}
$$

where $\alpha_t := f(t)$ and $\sigma_t := b\left(1 - \exp\left(-\frac{g(t)}{b}\right)\right)$. Hence, in order to minimize the CFM loss, we only need to sample $t \sim \mathcal{U}[0, 1]$, $x_0 \sim X_0$, and $u \sim \mathcal{U}(-1, 1)$. Note the similarity between the MMD path (31) and the FM/diffusion path (7); by choosing $b = 1$, $f(t) := 1 - t$ and $g(t) := -\log(1 - t)$ it follows $\alpha(t) = 1 - t$, $\sigma(t) = t$, and we obtain in (32) the FM-velocity along the trajectory (30), where the Gaussian noise $z \sim \mathcal{N}(0, 1)$ is just replaced by a uniform noise $u \sim \mathcal{U}(-1, 1)$.

## C. IMM with Quantile Interpolants

In this section, we want to demonstrate how the IMM framework proposed in (Zhou et al., 2025) can be realized by our quantile approach.

The general idea of consistency models is to predict the jumps from a process $Z_t$ to the target $X_0$, while factoring in the *consistency* of the trajectory of $Z_t$ via $Z_s$, $0 < s < t$. In FM, this consistency of the flow is usually neglected as only single points on the FM paths are sampled. Also, consistency models as one-step or multistep samplers usually are in no need of velocity fields.

Note that in the following – for notational simplicity – we consider the one-dimensional case $X_0, Z_t \in \mathbb{R}$ where we can employ quantile functions. By combining the 1D components into a multivariate model $\mathbf{X}_0 = (X_0^1, ..., X_0^d)$, $\mathbf{Z}_t = (Z_t^1, ..., Z_t^d)$, the results of this chapter trivially extend to $\mathbb{R}^d$.

Recall our definition of the *quantile process*

$$
Z_t = f(t)X_0 + Q_{g(t)}(U), \quad U \sim \mathcal{U}(0, 1), \, t \in [0, 1]. \tag{33}
$$

and the *quantile interpolants*

$$
I_{s,t}(x, y) = f(s)x + Q_{g(s)}\big(R_{g(t)}(y - f(t)x)\big), \quad s, t \in [0, 1]. \tag{34}
$$

Note that by the assumptions (5) it holds $Z_0 = X_0$ and $Z_1 = Q_1(U)$.

By the following remark, our quantile interpolants generalize the interpolants used in Denoising Diffusion Implicit Models (DDIM).

*Remark* C.1 (Relation to DDIM). The interpolants used in Denoising Diffusion Implicit Models (DDIMs) (Song et al., 2020) are given by

$$\text{DDIM}_{s,t}(x, y) := \left(\alpha_s - \frac{\sigma_s}{\sigma_t}\alpha_t\right)x + \frac{\sigma_s}{\sigma_t}y. \tag{35}$$

Now let $f(t) := 1 - t$, $g(t) := t^2$ and let $Q_t$ be the quantile of the law of a standard Brownian motion $W_t$.

First we obtain

$$Q_{g(t)}(p) = Q_{t^2}(p) = Q_{\mathcal{N}(0,t^2)}(p) = t\sqrt{2}\,\text{erf}^{-1}(2p - 1) = t\,Q_{\mathcal{N}(0,1)}(p), \quad p \in (0, 1),$$

with the *error function* erf. Hence, (33) exactly becomes (not only in distribution)

$$Z_t = (1 - t)Y_0 + t\,Q_{\mathcal{N}(0,1)}(U) = (1 - t)Y_0 + tZ,$$

where $Z := Q_{\mathcal{N}(0,1)}(U) \sim \mathcal{N}(0, 1)$, i.e. the components of (7) with the choice $\alpha_t = 1 - t$, $\sigma_t = t$. Furthermore, since $R_{t^2}(z) = R_{\mathcal{N}(0,t^2)}(z) = \frac{1}{2}(1 + \text{erf}\left(\frac{z}{t\sqrt{2}}\right))$, the quantile interpolant (14) reads as

$$I_{s,t}(x, y) = (1 - s)x + s\sqrt{2}\,\text{erf}^{-1}\left(\text{erf}\left(\frac{y - (1 - t)x}{t\sqrt{2}}\right)\right) = (1 - s)x + \frac{s}{t}(y - (1 - t)x)$$

$$= ((1 - s) - \frac{s}{t}(1 - t))x + \frac{s}{t}y.$$

which is exactly $\text{DDIM}_{s,t}(x, y)$ in (35) with $\alpha_t = f(t)$ and $\sigma_t^2 = g(t)$. $\diamond$

Exactly as the DDIM interpolants, our quantile interpolants (34) satisfy the following crucial interpolation properties.

**Proposition C.2** (a.k.a Proposition 4.1). *For all $x, y \in \mathbb{R}$ and all $s, r, t \in [0, 1]$, it holds*

$$I_{0,t}(x, y) = x, \quad I_{t,t}(x, y) = y, \tag{36}$$

*and*

$$I_{s,r}(x, I_{r,t}(x, y)) = I_{s,t}(x, y).$$

*Furthermore, inserting the quantile process (13) yields*

$$I_{s,t}(Z_0, Z_t) = Z_s. \tag{37}$$

*Proof.* By assumptions it holds

$$I_{0,t}(x, y) = f(0)x + Q_{g(0)}\left(R_{g(t)}(y - f(t)x)\right) = x,$$

and

$$I_{t,t}(x, y) = f(t)x + Q_{g(t)}\left(R_{g(t)}(y - f(t)x)\right) = y.$$

Furthermore, it holds the interpolation/consistency property

$$I_{s,r}(x, I_{r,t}(x, y)) = f(s)x + Q_{g(s)}\left(R_{g(r)}(I_{r,t}(x, y) - f(r)x)\right)$$
$$= f(s)x + Q_{g(s)}\left(R_{g(r)}(f(r)x + Q_{g(r)}(R_{g(t)}(y - f(t)x)) - f(r)x)\right)$$
$$= f(s)x + Q_{g(s)}\left(R_{g(t)}(y - f(t)x)\right)$$
$$= I_{s,t}(x, y)$$

for all $x, y \in \mathbb{R}$. Also note that inserting the random variables $Z_0, Z_t$ yields

$$I_{s,t}(Z_0, Z_t) = f(s)Z_0 + Q_{g(s)}\left(R_{g(t)}(Z_t - f(t)Z_0)\right)$$
$$= f(s)Z_0 + Q_{g(s)}(U)$$
$$= Z_s.$$

This finishes the proof. $\square$

Proposition C.2 represents the key observation which allows us to utilize our quantile process (33) in the IMM framework the same way as (Zhou et al., 2025) employ the DDIM interpolants (35):

For this, let us now recall the basic idea of inductive moment matching and the corresponding loss functions. Let us distinguish between real numbers written in small letters ($x_0, u, z_t \in \mathbb{R}$) and random variables written with capital letters ($X_0, U, Z_t, \ldots$). We assume that the probability distributions have densities:

| $\mathrm{Law}(X_0)$ | $\mathrm{Law}(Z_t)$ | $\mathrm{Law}(Z_s|X_0 = x_0, Z_t = z_t)$ | $\mathrm{Law}(Z_t|X_0 = x_0, U = u)$ | $\mathrm{Law}(X_0|Z_t = z_t)$ |
|---|---|---|---|---|
| $\rho_0(x_0)$ | $\rho_t(z_t)$ | $\rho_{s|0,t}(z_s|x_0, z_t)$ | $\rho_{t|0,1}(z_t|x_0, u)$ | $\rho_{0|t}(x_0|z_t)$ |

Note that by (37) we have $\rho_{s|0,t}(z_s|x_0, z_t) = \mathrm{Law}(I_{s,t}(x_0, z_t))(z_s) = \delta(z_s - I_{s,t}(x_0, z_t))$, hence sampling from $\rho_{s|0,t}(z_s|x_0, z_t)$ is just applying $I_{s,t}(x_0, z_t)$. Similarly, sampling from $\rho_{t|0,1}(z_t|x_0, u)$ is just evaluating $I_{t,1}(x_0, Q_1(u))$.

The following proposition follows directly from Proposition C.2 as in (Zhou et al., 2025). It is essential for deriving the appropriate loss functions.

**Proposition C.3.** *For all $0 \leq s \leq r \leq t \leq 1$, the quantile interpolant (34) is self-consistent, i.e.*

$$\rho_{s|0,t}(z_s|x_0, z_t) = \int_{\mathbb{R}} \rho_{s|0,r}(z_s|x_0, z_r)\, \rho_{r|0,t}(z_r|x_0, z_t)\, \mathrm{d}z_r,$$

*and the quantile process (33) is marginal preserving, i.e.*

$$\rho_s(z_s) = \mathbb{E}_{z_t \sim \rho_t, x_0 \sim \rho_{0|t}(\cdot|z_t)} \left[ \rho_{s|0,t}(z_s|x_0, z_t) \right].$$

**Learning.** The conditional probability $\rho_{0|t}(\cdot|z_t)$ is now approximated by a network $p^\theta_{s,t,z_t}$ where the parameter $s$ describes the dependence on $\rho_s$ such that

$$\rho_s \approx \mathbb{E}_{z_t \sim \rho_t, x_0 \sim p^\theta_{s,t,z_t}} \left[ \rho_{s|0,t}(\cdot|x_0, z_t) \right] =: p^\theta(s,t). \tag{38}$$

Then it is proposed in (Zhou et al., 2025) to minimize the so-called *naïve objective*

$$\mathcal{L}_{\mathrm{naive}}(\theta) := \mathbb{E}_{s,t} \left[ D(\rho_s, p^\theta(s,t)) \right], \tag{39}$$

with an appropriate metric $D$, e.g. MMD. The procedure is now as follows: starting in a sample $x_0$ from $X_0$, we can sample $z_s, z_t$ from $Z_s, Z_t$ by (33), respectively; then given $z_t$ we sample $\tilde{x}_0$ from $p^\theta_{s,t,z_t}$, and finally we can evaluate $\tilde{z}_s = I(\tilde{x}_0, z_t)$ from (37), which is then compared with $z_s$.

**Inference.** The following iterative multi-step sampling can be applied: for chosen decreasing $t_k \in (0,1]$, $k = 0, \ldots, T$ with $t_0 = 1$, starting with $x_0^{(0)} \sim p^\theta_{0,1,z_1}$, we compute

$$z_{t_k} = I_{t_k, t_{k-1}}\left(x_0^{(k-1)}, z_{t_{k-1}}\right), \quad x_0^{(k)} \sim p^\theta_{0,t_k,z_{t_k}}, \quad k = 1, \ldots, T.$$

Although for marginal-preserving interpolants, a minimizer of $\mathcal{L}_{\mathrm{naive}}$ exists with minimum 0, the authors of (Zhou et al., 2025) object that directly optimizing (39) faces practical difficulties when $t$ is far away from $s$. Instead, they propose to apply the following "inductive bootstrapping" technique:

**Bootstrapping.** Instead of minimizing (39), we consider the *general objective*

$$\mathcal{L}_{\mathrm{general}}(\theta) := \mathbb{E}_{s,t} \left[ w(s,t) \mathrm{MMD}^2_K(p^{\theta_{n-1}}(s,r), p^{\theta_n}(s,t)) \right], \tag{40}$$

with a weighting function $w(s,t)$ to be chosen. The kernel $K$ of the squared MMD distance can be chosen as e.g. the (time-dependent) Laplace kernel. Importantly, the value $r$ is chosen to be a function $r = r_{s,t} \in [s,t]$ being "close to $t$" and fulfilling a suitable monotonicity property.

Let us assume the simplest case $r_{s,t} := \max\{s, t - \varepsilon\}$ with a small fixed $\varepsilon > 0$ and hereby demonstrate the bootstrapping technique: Fix $s \in [0,1]$. Then, it holds for all $t \in [s, s + \varepsilon]$ that $r_{s,s} = s$. By the definition (38) and property (36), it holds (independently of $\theta$) that $p^\theta(s,s)(z_s) = \rho_s(z_s)$. Hence, minimizing (40) in the first step $n = 1$ yields

$$0 = \mathrm{MMD}^2_K(p^{\theta_0}(s,s), p^{\theta_1}(s, t_1)) = \mathrm{MMD}^2_K(\rho_s, p^{\theta_1}(s, t_1)) \quad \text{for all } t_1 \in [s, s + \varepsilon].$$

In the second step $n = 2$, it holds for all $t_2 \in [s, s + 2\varepsilon]$ that $r_{s,t_2} \in [s, s + \varepsilon]$. Hence, minimizing (40) in the second step yields, together with the first step,

$$0 = \mathrm{MMD}_K^2(p^{\theta_1}(s, r_{s,t_2}),\, p^{\theta_2}(s, t_2)) = \mathrm{MMD}_K^2(\rho_s, p^{\theta_2}(s, t_2)) \quad \text{for all } t_2 \in [s, s + 2\varepsilon].$$

Thus, for the number of steps $n \to \infty$, it holds $0 = \mathrm{MMD}_K^2(\rho_s, p^{\theta_n}(s, t_n))$ even for the entire interval $t_n \in [s, 1]$. Hence, minimizing the general objective (40) with a large number of steps eventually minimizes the naïve objective (39), see Theorem 1 (Zhou et al., 2025) for more details.

## D. Radially Symmetric Processes

While our main construction relies on independent one-dimensional processes across coordinates (Section 3), we can alternatively build multi-dimensional processes using radial symmetry. This approach, sometimes called *polar factorization*, decomposes a random vector as the product of its radius and direction.

Let $x \in \mathbb{R}^d$, write $r = \|x\|$ and $\omega = x/\|x\| \in \mathbb{S}^{d-1}$. A vector field is called *radial* if

$$v_t(x) = v_t^R(r)\,\omega,$$

where $v_t^R : [0, \infty) \to \mathbb{R}$ is its radial component. Likewise, a time-dependent probability density $\rho_t : \mathbb{R}^d \to \mathbb{R}$ is *radially symmetric* if there exists $\bar{\rho}_t : [0, \infty) \to \mathbb{R}$ such that

$$\rho_t(x) = \bar{\rho}_t(\|x\|) \quad \text{for all } x \in \mathbb{R}^d.$$

Define the radial mass density

$$f_R(t, r) := \int_{\mathbb{S}^{d-1}} \rho_t(r\omega)\, r^{d-1}\, \mathrm{d}\sigma(\omega) = |\mathbb{S}^{d-1}|\, r^{d-1}\, \bar{\rho}_t(r), \tag{41}$$

where $\sigma$ denotes the surface measure on $\mathbb{S}^{d-1}$. If $\mathbf{X} \in \mathbb{R}^d$ has radially symmetric density $\rho$, then $f_R$ is precisely the probability density of the radius $\|\mathbf{X}\|$.

**Lemma D.1.** *Let $\rho_t$ be a smooth, radially symmetric density on $\mathbb{R}^d$, and let*

$$v_t(x) = v_t^R(r)\,\frac{x}{r}$$

*be a smooth radial velocity field. Then, the pair $(\rho_t, v_t)$ satisfies the d-dimensional continuity equation (1) if and only if $(f_R(t, \cdot), v_t^R)$ fulfills the one-dimensional flux equation for $r > 0$:*

$$\partial_t f_R(t, r) + \partial_r\big[f_R(t, r)\, v_t^R(r)\big] = 0, \tag{42}$$

*where $f_R$ is defined in (41).*

*Proof.* We compute the divergence in Cartesian coordinates:

$$\nabla_x \cdot \big[\bar{\rho}_t(r)\, v_t^R(r)\, \tfrac{x}{r}\big] = \sum_{i=1}^{d} \partial_{x_i}\Big(\bar{\rho}_t(r)\, v_t^R(r)\, \tfrac{x_i}{r}\Big).$$

Since $\bar{\rho}_t$ and $v_t^R$ depend only on $r = \|x\|$, and $\partial_{x_i} r = x_i/r$, the product rule gives

$$\partial_{x_i}\Big(\bar{\rho}_t\, v_t^R\, \tfrac{x_i}{r}\Big) = \partial_r(\bar{\rho}_t\, v_t^R)\,\frac{x_i^2}{r^2} + \bar{\rho}_t\, v_t^R\,\frac{1}{r} - \bar{\rho}_t\, v_t^R\,\frac{x_i^2}{r^3}.$$

Summing over $i$ and using $\sum_{i=1}^{d} x_i^2 = r^2$ yields

$$\nabla_x \cdot (\bar{\rho}_t\, v_t) = \partial_r(\bar{\rho}_t\, v_t^R) + \frac{d-1}{r}\,\bar{\rho}_t\, v_t^R = r^{-(d-1)}\, \partial_r\big(r^{d-1}\, \bar{\rho}_t\, v_t^R\big).$$

Hence the continuity equation (1) is equivalent to

$$\partial_t \bar{\rho}_t + \frac{1}{r^{d-1}}\, \partial_r\big(r^{d-1} \bar{\rho}_t\, v_t^R\big) = 0.$$

Multiplying by $|\mathbb{S}^{d-1}|\, r^{d-1}$ shows the equivalence to (42). $\qquad\square$

**Definition D.2** (Radial Processes). Let $(R_t)_t$ be a non-negative one-dimensional stochastic process with $R_0 = 0$, probability density $p_t^R$, and velocity field $v_t^R$ satisfying the one-dimensional continuity equation (10). Let $\mathbf{U} \sim \mathrm{Unif}(\mathbb{S}^{d-1})$ be uniformly distributed on the unit sphere, independent of $R_t$. The *radial process w.r.t.* $(R_t)_t$ in $\mathbb{R}^d$ is defined by the polar factorization

$$\mathbf{Y}_t = R_t \, \mathbf{U}.$$

Since $\|\mathbf{Y}_t\| = |R_t|$, we have $f_R(t, r) = p_t^R(r)$ for all $r > 0$.

The denotation of the velocity field of the process $(R_t)_t$ as $v_t^R$ is justified by the following proposition.

**Proposition D.3** (Radial Continuity). *Let $\mu_t$ denote the law of $\mathbf{Y}_t$ in Definition D.2, with radially symmetric density $\rho_t(x) = \bar{\rho}_t(\|x\|)$. Let $p_t^R$ and $v_t^R$ be given as in Definition D.2.*

*Then, the velocity field*

$$v_t(x) := v_t^R(\|x\|) \, \frac{x}{\|x\|}$$

*satisfies the $d$-dimensional continuity equation (1) together with $\mu_t$. In other words, the radial process $\mathbf{Y}_t$ admits a radial velocity field $v_t$ with radial component $v_t^R$ given by $R_t$.*

*Proof.* By Lemma D.1, it suffices to verify (42). But we have

$$\partial_t f_R(t, r) + \partial_r[f_R(t, r) \, v_t^R(r)] = \left[\partial_t p_t^R(r) + \partial_r(p_t^R \, v_t^R)(r)\right] = 0,$$

since $(p_t^R, v_t^R)$ satisfies the one-dimensional continuity equation (10) by definition. $\square$

*Remark* D.4 (Connection to Mean-Reverting Processes and Learning). The radial construction integrates with the mean-reverting framework from Section 2.4. Given $\mathbf{X}_0 \sim \mu_0$ and scheduling functions $f, g$ satisfying (5), we define

$$\mathbf{X}_t = f(t) \, \mathbf{X}_0 + R_{g(t)} \, \mathbf{U},$$

where $R_t$ is a radial process and $\mathbf{U} \sim \mathrm{Unif}(\mathbb{S}^{d-1})$ is independent of $\mathbf{X}_0$. For instance, using the Kac process as the underlying one-dimensional radial process yields a radially symmetric alternative to the componentwise construction.

For the scaled latent case, choosing $f(t) = 1 - t$, $g(t) = t$, and $R_t = t R$ where $R \sim \chi_d$ follows the chi distribution with $d$ degrees of freedom (i.e., $R = \|\mathbf{Z}\|$ for $\mathbf{Z} \sim \mathcal{N}(0, I_d)$), we recover the standard Gaussian latent $\mathbf{Y}_1 \sim \mathcal{N}(0, I_d)$.

The radial construction naturally extends to the quantile learning framework of Section 4. We can parametrize the radial distribution via its quantile function $Q_R : (0, 1) \to [0, \infty)$, where $R_t = t \, Q_R(U)$ with $U \sim \mathcal{U}(0, 1)$. Following Section 5, we learn $Q_R$, which can also be combined with velocity field training analogously to (16). This provides a data-adapted isotropic latent, in contrast to the fully independent coordinate-wise adaptation of our main method.

# E. Adapting Noise to Data

## E.1. Counterexample: Marginal Product

For the measure

$$\mu = \tfrac{1}{2}\delta_{(1,1)} + \tfrac{1}{2}\delta_{(-1,-1)} \in \mathcal{P}_2(\mathbb{R}^2), \qquad \mu_{\mathrm{marg}} = \left(\tfrac{1}{2}\delta_{-1} + \tfrac{1}{2}\delta_1\right) \otimes \left(\tfrac{1}{2}\delta_{-1} + \tfrac{1}{2}\delta_1\right),$$

one has $W_2^2(\mu, \mu_{\mathrm{marg}}) = 2$, whereas for

$$\nu_\alpha = \left(\tfrac{1}{2}\delta_{-\alpha} + \tfrac{1}{2}\delta_\alpha\right) \otimes \left(\tfrac{1}{2}\delta_{-\alpha} + \tfrac{1}{2}\delta_\alpha\right)$$

it holds $W_2^2(\mu, \nu_\alpha) = 2\big(1 - \alpha + \alpha^2\big) = 1.5$ for $\alpha = 0.5$. Thus the $W_2$–closest independent latent may contract or expand the marginals to partially account for correlations it cannot represent.

### E.2. Details on the Architecture of the Learned Quantiles

We implement each one–dimensional quantile function with rational–quadratic splines (RQS) (Gregory & Delbourgo, 1982; Durkan et al., 2019). We explored several ways to map $u \in (0,1)$ into the spline input; the two variants below consistently performed well and are used in our experiments. For every coordinate $i$, we write

$$Q_\phi^i(u) = S_\phi^i(\psi(u)), \qquad u \in (0,1),$$

where $S_\phi^i : \mathbb{R} \to \mathbb{R}$ is a strictly increasing RQS with an interior knot interval $(-B, B)$ (with $K$ bins) and linear tails outside $\pm B$ that are $C^1$-matched at the boundaries. The two settings differ only in the "activation" $\psi$:

$$\text{(A) Logit: } \psi(u) = \text{logit}(u), \qquad \text{(B) Affine: } \psi(u) = \alpha_B(u) = B(2u - 1).$$

Thus, both (A) and (B) share exactly the same spline $S_\phi^i$ architecture—including the bounded interior $(-B, B)$ and slope-matched linear tails—and differ only in how $(0,1)$ is mapped into the spline's input. In (A), $\psi(u) \in \mathbb{R}$ and the linear tails of $S_\phi^i$ are used whenever $|\text{logit}(u)| > B$; in (B), $\psi(u) \in (-B, B)$ so the forward pass never touches the tails (they remain important for invertibility and out-of-range evaluation).

**Parameterization and Constraints.** Each spline $S_\phi^i$ is parameterized by raw bin widths, heights, and knot slopes. We pass these raw parameters through softplus, normalize widths and heights to sum to one (scaled to the domain span $2B$ and the learned range span, respectively), and add a small constant $s_{\min} > 0$ to each slope to enforce a positive lower bound. The linear tail slopes (left/right) are learned in the same way and are chosen so that both function value and slope agree at $\pm B$. These constraints guarantee strict monotonicity, hence $Q_\phi^i$ is strictly increasing on $(0,1)$ under both (A) and (B). Closed-form formulas for the spline pieces and their (log-)derivatives are available; by the chain rule,

$$\frac{d}{du} Q_\phi^i(u) = S_\phi^{i\prime}(\psi(u))\, \psi'(u), \quad \text{with} \quad \psi'(u) = \begin{cases} \frac{1}{u(1-u)} & \text{for (A)}, \\ 2B & \text{for (B)}. \end{cases}$$

**Per-Component Affine Wrapper (Scale/Bias).** After computing $Q_\phi^i(u)$, we add a tiny affine head per coordinate:

$$\tilde{Q}_\phi^i(u) = s_i\, Q_\phi^i(u) + b_i, \qquad s_i = \text{softplus}(\log \alpha_i), \quad b_i = \beta_i,$$

where $\alpha_i > 0$ and $\beta_i \in \mathbb{R}$ are learned per component. Using $\text{softplus}(\log \alpha_i)$ keeps $s_i > 0$ with a convenient dynamic range; this preserves monotonicity and adds only one scale and one bias parameter per component.

**Regularization via Expected Negative Log–Jacobian** Let $Q_\phi : (0,1)^d \to \mathbb{R}^d$ be the componentwise map with affine heads, $Q_\phi(u) = (\tilde{Q}_\phi^1(u_1), \ldots, \tilde{Q}_\phi^d(u_d))$. Since the construction is per–coordinate, the Jacobian is diagonal with entries $\partial_{u_i} \tilde{Q}_\phi^i(u_i) > 0$. We regularize with the expected negative log–determinant of the Jacobian:

$$\mathcal{L}_{\text{reg}}(\phi) = \lambda_{\text{reg}}\, \mathbb{E}_{u \sim p_U}\big[-\log\det J_{Q_\phi}(u)\big]$$

$$= \lambda_{\text{reg}}\, \mathbb{E}_{u \sim p_U}\Big[-\sum_{i=1}^d \log\big(\partial_{u_i} \tilde{Q}_\phi^i(u_i)\big)\Big].$$

Here $p_U = \text{Unif}\big((0,1)^d\big)$. In practice, we evaluate the log–derivatives in closed form.

**Computational Efficiency and Scalability.** The quantile architecture is highly efficient in both computation and memory. Each component $i$ requires only $\mathcal{O}(K)$ parameters for the RQS (where $K$ is the number of bins) plus two affine parameters, totaling roughly $4K + 2$ parameters per dimension for typical implementations. For a $d$-dimensional problem, this yields $\mathcal{O}(d \cdot K)$ total parameters—negligible compared to modern UNet architectures which often contain millions of parameters. Forward evaluation of $Q_\phi(u)$ involves $d$ independent spline evaluations operating in parallel. The diagonal Jacobian structure means that both the determinant and its gradient reduce to $d$ independent scalar derivatives with analytical closed-form expressions which are fully parallelizable, avoiding expensive automatic differentiation of matrix operations.

In practice, as noted in Section 6, the computational overhead (on CIFAR10) during joint training is approximately $2.7\%$ and drops to $0.5\%$ after freezing the quantile. Furthermore we only used 300k parameters for the quantile in contrast to 35M for the U-Net, making the approach highly scalable to high-dimensional problems. The strict monotonicity constraints and bounded parameterization (via softplus and normalization) ensure numerical stability throughout training, and we observed no instabilities across our experiments spanning dimensions from $d = 2$ to $d = 3072$ (CIFAR-10).

# F. Experiment Details

## F.1. Toy Target Distributions

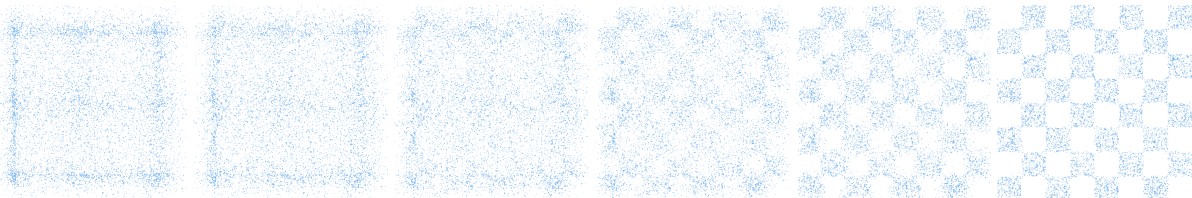

*Figure 11.* A generated sample path from the learned quantile latent to the checkerboard. The adapted latent (left) is already close to the target distribution.

We use three standard challenging low-dimensional distributions: Neal's funnel, a $3 \times 3$ Gaussian mixture, and a checkerboard.

**Funnel.**  For the toy illustration in Figure 2, we work with the dataset known as Neal's Funnel (Neal, 2003). The distribution of Neal's funnel is defined as follows:

$$p(x_1, x_2) = \mathcal{N}(x_1; 0, 3) \, \mathcal{N}(x_2; 0, \exp(x_1/2)).$$

**Grid Gaussian Mixture.**  We give more details about the mixture of Gaussian we consider in our experiment. It is designed in a grid pattern in $[-1, 1]^2$, as follows:

$$\sum_{i=1}^{9} w_i \cdot \mathcal{N}(\mu_i, \sigma^2 I_2),$$

where $(w_i)_{i=1}^{9} = (0.01, 0.1, 0.3, 0.2, 0.02, 0.15, 0.02, 0.15, 0.05)$, $\mu_i = (\mu_1, \mu_2)$ with $\mu_1 = (i \bmod 3) - 1$, $\mu_2 = \lfloor \frac{i}{3} \rfloor - 1$, and $\sigma = 0.025$.

**Checkerboard.**  Fix $\ell < h$ and domain $\Omega = [\ell, h]^2$. Define the support

$$\mathcal{S} = \big\{ (x, y) \in \Omega : \ \lfloor x \rfloor + \lfloor y \rfloor \text{ is even} \big\}.$$

The checkerboard distribution is uniform on $\mathcal{S}$ and zero elsewhere:

$$p_{\text{Checker}}(x, y) = \begin{cases} \dfrac{1}{\text{area}(\mathcal{S})}, & (x, y) \in \mathcal{S}, \\ 0, & \text{otherwise.} \end{cases}$$

For integer $\ell, h$ with even side length (e.g. $\ell = -4, h = 4$), exactly half of $\Omega$ is active, hence

$$p_{\text{Checker}}(x, y) = \frac{2}{(h - \ell)^2} \, \mathbf{1}_{\mathcal{S}}(x, y).$$

## F.2. Loss Implementation

For training, the minibatch OT is computed empirically as follows: draw a minibatch $\{\mathbf{x}_0^{(i)}\}_{i=1}^{B} \sim \mu_0$ and $\{\mathbf{u}^{(j)}\}_{j=1}^{B} \sim \mathcal{U}([0,1]^d)$, set $\mathbf{y}^{(j)} = \mathbf{Q}_\phi(\mathbf{u}^{(j)})$, and define the empirical measures

$$\hat{\mu}_0^B = \tfrac{1}{B} \sum_{i=1}^{B} \delta_{\mathbf{x}_0^{(i)}}, \qquad \hat{\nu}_\phi^B = \tfrac{1}{B} \sum_{j=1}^{B} \delta_{\mathbf{y}^{(j)}}.$$

The minibatch quantile alignment objective is

$$\widehat{\mathcal{L}}_{\text{AN}}(\phi) = W_2^2 \big( \hat{\mu}_0^B, \hat{\nu}_\phi^B \big),$$

and gradients backpropagate through $\mathbf{y}^{(j)} = \mathbf{Q}_\phi(\mathbf{u}^{(j)})$. Let $T : \{1, \ldots, B\} \to \{1, \ldots, B\}$ denote the optimal assignment that minimizes $\sum_{i=1}^{B} \|\mathbf{x}_0^{(i)} - \mathbf{y}^{(T(i))}\|_2^2$, and define its inverse $P(j) = i$ such that $T(i) = j$. We use the conditional flow path $\mathbf{x}_t^{(j)} = (1 - t_j)\mathbf{x}_0^{(P(j))} + t_j\,\mathbf{y}^{(j)}$, $j = 1, \ldots, B$, with $t_j \sim \mathcal{U}(0, 1)$. The target velocity is $\mathbf{y}^{(j)} - \mathbf{x}_0^{(P(j))}$, we apply a stop-gradient operator $\mathrm{sg}(\cdot)$ to this target in the flow matching loss. This prevents gradients from the velocity model from flowing back through the quantile function in this term, ensuring that $\mathbf{Q}_\phi$ is updated primarily through $\widehat{\mathcal{L}}_{\mathrm{AN}}$, while $v^\theta$ learns to match the transport directions defined by the current quantile map. Note however the stop gradient operation only slightly stabilizes training, we can train with full gradients as well. We optimize the empirical version

$$\widehat{\mathcal{L}}_{\mathrm{CFM}}(\theta, \phi) = \frac{1}{B}\sum_{j=1}^{B} \left\| v^\theta\big(\mathbf{x}_t^{(j)}, t_j\big) - \mathrm{sg}\big(\mathbf{y}^{(j)} - \mathbf{x}_0^{(P(j))}\big) \right\|_2^2, \quad \widehat{\mathcal{L}}(\theta, \phi) = \widehat{\mathcal{L}}_{\mathrm{CFM}}(\theta, \phi) + \lambda\,\widehat{\mathcal{L}}_{\mathrm{AN}}(\phi).$$

### F.3. Implementation Details

We support baseline flow matching, optional quantile pretraining, and joint quantile+velocity optimisation. Pretraining fits the RQS transport before optionally freezing it; joint training updates both modules simultaneously. Once the quantile learning rate decays to zero we freeze its weights and continue optimising the velocity field only.

The coupling plans are calculated using the Python Optimal Transport package (Flamary et al., 2021). For inference simulate the corresponding ODEs using the torchdiffeq (Chen, 2018) package. For all models we only used the batch size 128 and learning rate $2e-4$ for the velocities. We use Adam (Kingma & Ba, 2015) as the optimizer. The quantiles are parameterised by rational-quadratic splines as described in E.2, we set the minimum bin width and height to $1e-3$ and the minimum slope to $1e-5$. We could in principle stack multiple RQS layers, however for all of our experiments we use one layer.

### F.4. Experiment Setup: Synthetic Examples

All models include a sinusoidal time embedding and SiLU activation functions.

**Funnel.** For all models we used 3 hidden layers with width 64. We used a batch size of 128, a learning rate of $2e-4$ and exponential moving average on the network weights of 0.999. The baselines were trained for 200,000 iterations. Since there is a very high variance when sampling from the funnel, we pretrain our quantiles and use the frozen quantiles during flow matching. We trained our quantile for 50,000 steps and to compensate we trained our velocity for only 100,000 steps. For the RQS we chose logit activation, 32 bins and a bound of 500.

**Grid Gaussian Mixture and Checker.** The quantiles were trained for the first 20,000 steps, after which the learning rate was linearly decayed to 0 by step 25,000. For both datasets, we trained the velocity model with 3 layers and a hidden width of 256 for 100,000 steps. Furthermore we used sinusoidal positional embeddings for the checkerboard. For the RQS we chose logit activation, 32 bins with a bound of 50.

### F.5. Experiment Setup: Image Experiments

For both image datasets, we adapt the U-Net from (Nichol & Dhariwal, 2021) to parametrize our velocity field.

**MNIST.** For the MNIST dataset we use the U-Net with channel multipliers $(1, 2, 4)$, two residual blocks per resolution, attention at $7 \times 7$, and 1 attention head. We clip the gradient norm to 1 and use exponential moving averaging with a decay of 0.99. We test three configurations with base widths of 8, 16, and 32 channels. For these ablation runs, we use quantile loss weight $\lambda = 1.0$, regularization parameter $\beta = 0.1$, and rational quadratic spline with bounded activation, 16 bins and bound 3.0. The quantiles were trained for the first 20,000 steps, after which the learning rate was linearly decayed to 0 over the next 10,000 steps. The images in Figure 5 were generated using our 32 channel configuration.

**CIFAR.** We use exactly the same U-Net setup from (Tong et al., 2024). We clip the gradient norm to 1 and use exponential moving averaging with a decay of 0.9999. To evaluate our results, we use the Fréchet inception distance (FID) (Heusel et al., 2017). The quantiles were trained for the first 50,000 steps, after which the learning rate was linearly decayed to 0 by step 55,000. We used $\lambda = 1$ and varied $\beta$. For the RQS we used logit activation, 32 bins and a bound of 25.

CIFAR-10 inputs are normalized to $[-1, 1]$ with random horizontal flips.

## F.6. Experiment Setup: Weather Experiments

**HRRR64.** For the HRRR64-Mini dataset, we use a U-Net with channel multipliers $(1, 2, 2, 2)$, two residual blocks per resolution, attention at $32 \times 32$, and 4 attention heads with 64 channels per head. The base width is 128 channels, with dropout 0.1 and bf16 mixed precision. We clip the gradient norm to 1 and use exponential moving averaging with decay 0.99. Training uses Adam with learning rate $2 \times 10^{-4}$, 5,000 warmup steps, batch size 64, and minibatch optimal-transport coupling between data and latents, for a total of 100,000 steps. For quantile training, we use loss weight $\lambda = 0.3$, regularization parameter $\beta = 1.0$, and a single rational-quadratic spline layer with a logit input transform, 32 bins, and bound 25.0. The quantiles are trained for the first 5,000 steps, after which the learning rate is linearly decayed to 0 over the next 2,500 steps. The Student-$t$ baseline uses $\nu = 4$ degrees of freedom with unit scale.

**Metrics.** Let $X$ denote the total-precipitation pixel value, and write $X_{\text{gen}}$ and $X_{\text{real}}$ for $X$ under the generative model and the real data respectively. We pool generated and real precipitation pixels across all images and pixel locations, denote their empirical distributions by $\hat{F}_{\text{gen}}$ and $\hat{F}_{\text{real}}$, and write $\hat{Q}_{\text{real}}(\tau) = \hat{F}_{\text{real}}^{-1}(\tau)$ for the empirical quantile of the real data. We fix the extreme threshold at the 99.9th percentile,

$$u_\tau = \hat{Q}_{\text{real}}(\tau), \qquad \tau = 0.999.$$

All metrics below are evaluated on the tp channel; lower is better.

The *extreme event frequency error* is the relative error of the tail exceedance probability,

$$\text{EEFE}(X_{\text{gen}}, X_{\text{real}}) = \frac{\left| \mathbb{P}(X_{\text{gen}} > u_\tau) - \mathbb{P}(X_{\text{real}} > u_\tau) \right|}{\mathbb{P}(X_{\text{real}} > u_\tau)},$$

and the *extreme event magnitude error* is the corresponding relative error of the tail conditional mean,

$$\text{EEME}(X_{\text{gen}}, X_{\text{real}}) = \frac{\left| \mathbb{E}[X_{\text{gen}} \mid X_{\text{gen}} > u_\tau] - \mathbb{E}[X_{\text{real}} \mid X_{\text{real}} > u_\tau] \right|}{\left| \mathbb{E}[X_{\text{real}} \mid X_{\text{real}} > u_\tau] \right|}.$$

For the *spectral distance*, let $\mathcal{F}[x](u, v)$ denote the 2-D DFT of a single realization $x \in \mathbb{R}^{H \times W}$. The isotropic (radially averaged) power spectrum of $x$ is the conditional expectation of $|\mathcal{F}[x]|^2$ over rings of constant radial wavenumber,

$$\hat{S}_x(k) = \mathbb{E}_{(u,v)}\left[ \left| \mathcal{F}[x](u, v) \right|^2 \mid (u, v) \in A_k \right], \qquad A_k = \left\{ (u, v) : \lfloor \sqrt{u^2 + v^2} + \tfrac{1}{2} \rfloor = k \right\},$$

i.e. the mean squared FFT magnitude over the unit-width annulus at integer radius $k$. Averaging $\hat{S}_x$ over the generated and real image sets gives $\hat{S}_{\text{gen}}$ and $\hat{S}_{\text{real}}$, and with $k_{\max} = \lfloor \min(H, W)/2 \rfloor$ and the DC bin excluded we report

$$\text{SD}(X_{\text{gen}}, X_{\text{real}}) = \frac{1}{k_{\max} - 1} \sum_{k=1}^{k_{\max}-1} \left| \log \hat{S}_{\text{gen}}(k) - \log \hat{S}_{\text{real}}(k) \right|,$$

with a small $\epsilon$ added inside each $\log$ for numerical stability.

The *tail KS distance* is the two-sample Kolmogorov–Smirnov statistic restricted to the tails of the real distribution. Letting $\mathcal{T}_+ = \{x : x > u_\tau\}$ and $\mathcal{T}_- = \{x : x < \hat{Q}_{\text{real}}(1 - \tau)\}$,

$$\text{TailKS}(X_{\text{gen}}, X_{\text{real}}) = \tfrac{1}{2} \sum_{\mathcal{T} \in \{\mathcal{T}_+, \mathcal{T}_-\}} \sup_{x \in \mathcal{T}} \left| \hat{F}_{\text{gen}}^{\mathcal{T}}(x) - \hat{F}_{\text{real}}^{\mathcal{T}}(x) \right|,$$

where $\hat{F}^{\mathcal{T}}$ is the empirical CDF restricted to the tail region $\mathcal{T}$.

Finally, the *kurtosis deviation* and *skewness deviation* are the relative errors of the Pearson kurtosis $\kappa = \mathbb{E}[(X - \mu)^4]/\sigma^4$ (Gaussian $\approx 3$) and the skewness $\gamma = \mathbb{E}[(X - \mu)^3]/\sigma^3$, with $\kappa_{\text{gen}}, \gamma_{\text{gen}}$ and $\kappa_{\text{real}}, \gamma_{\text{real}}$ computed from $X_{\text{gen}}$ and $X_{\text{real}}$ respectively,

$$\text{KD}(X_{\text{gen}}, X_{\text{real}}) = \left| 1 - \kappa_{\text{gen}}/\kappa_{\text{real}} \right|, \qquad \text{SkD}(X_{\text{gen}}, X_{\text{real}}) = \left| 1 - \gamma_{\text{gen}}/\gamma_{\text{real}} \right|.$$

All probabilities, expectations, quantiles, and CDFs are estimated from 20,000 generated samples versus the full HRRR64-Mini precipitation field.

# G. Further Experimental Results

In the following, we highlight additional experimental results that are not included in the main paper.

### G.1. Synthetic Datasets

For the funnel experiment in Section 6.2, we visualize the learned latent distribution $\mathbf{Q}_\phi$ in Figure 12. We draw one million samples to accurately depict the distribution, and color each sample by the norm of the corresponding target sample after solving the ODE. The learned latent exhibits substantially heavier tails than a Gaussian, yielding improved tail behavior compared to a standard Gaussian, as also illustrated in Figure 2.

For the checkerboard distribution (see Section F.1), the velocity-field network converges in substantially fewer training iterations. After 20k training steps, the generated samples match the target distribution much more closely, see Figure 13.

In figure 14 we consider what happens when training the latent purely using the contribution from the FM loss, i.e setting $\lambda = 0$.

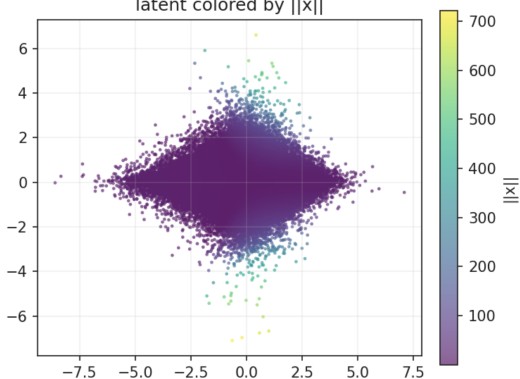

*Figure 12.* Samples (1M) from our learned latent of the funnel distribution. Color shows norm of the endpoint sample after solving the reverse ODE.

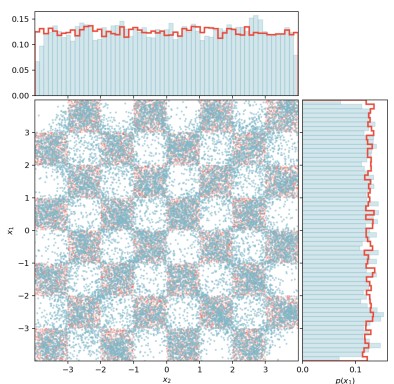
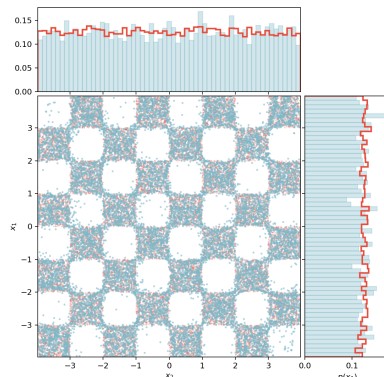

*Figure 13.* Flow Matching with optimal coupling using Gaussian noise (left) and our learned noise (right) after **20k** training steps with identical parameters. Starting in the learned noise the model produces much better samples. Generated samples are shown in blue, and ground-truth samples in red.

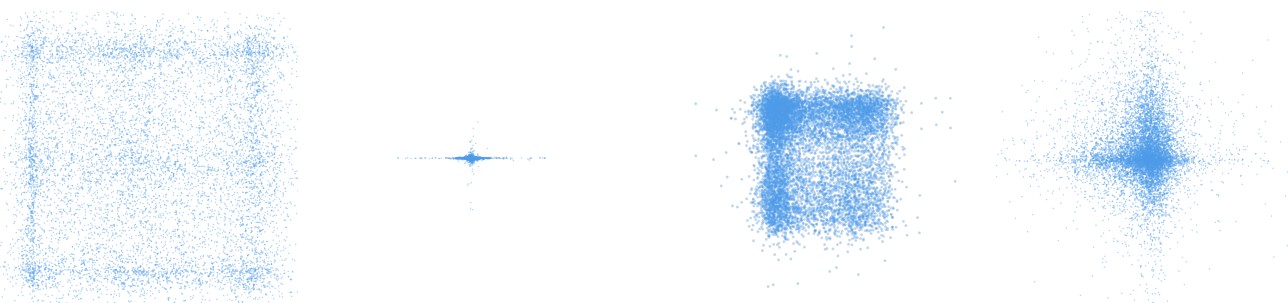

*(a)* Checkerboard (cf. Figure 1). Left: $\lambda = 50$, Right: $\lambda = 0$.   *(b)* Gaussian mixture (cf. Figure 3). Left: $\lambda = 50$, Right: $\lambda = 0$.

*Figure 14.* Effect of the regularization weight $\lambda$ on learned latent distributions for the checkerboard and GMM target distributions.

### G.2. Image Data Set: MNIST

For MNIST, Figure 15 visualizes the learned latent distribution $\mathbf{Q}_\phi$ at different pixel locations (e.g., the top-left corner, the center, and the mid-right region). The learned latent matches the correct mean, while the increased variance in the top-right region is induced by the differential-entropy regularization (see Section 6.3). We again plot FID over training steps for the capacity-constrained networks, and additionally report the corresponding parameter counts in Figure 16.

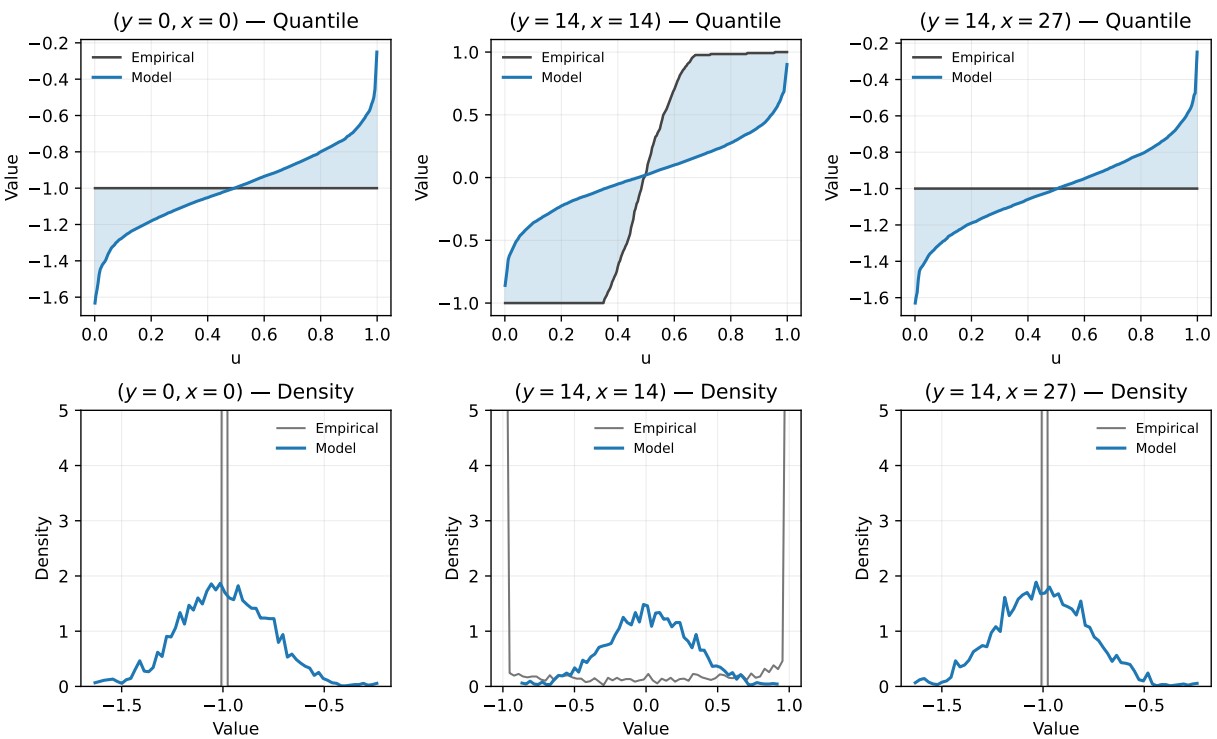

*Figure 15.* Comparison of the empirical and learned probability density functions and their quantile functions at different pixel locations $(y, x)$, averaged over images from the MNIST dataset. The blue area illustrates the difference between the quantiles, corresponding to the one-dimensional Wasserstein distance; see Eq. 4.1.

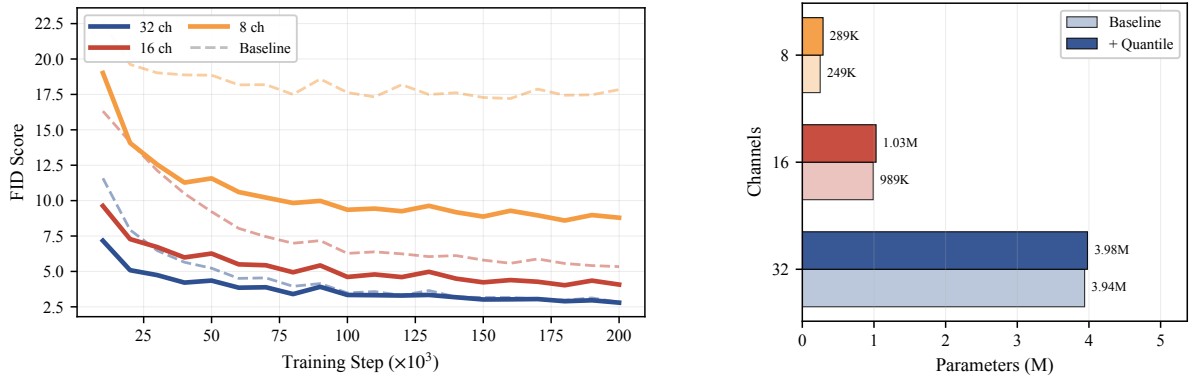

*Figure 16.* Ablation study over capacity of the U-Net for sampling from the MNIST dataset. The FID curves show that our method achieves significantly lower FIDs for lower capacities. Note the difference in parameters is approximately 40k.

## G.3. Image Dataset: CIFAR10

In Figure 17, we present the full ablation over the regularization parameter (see Section 6.3). Most regularization settings outperform the standard Gaussian noise distribution.

Figure 18 summarizes the training dynamics over iterations, including the Wasserstein distance, the differential-entropy regularization term, and the gradient norms of both the quantile and velocity-field networks.

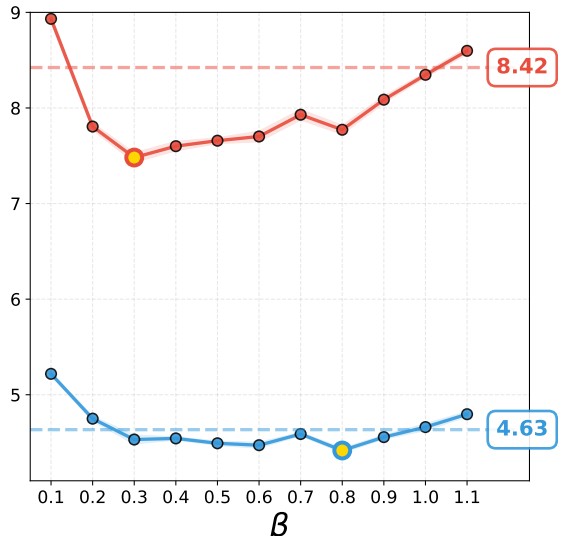

| $\beta$ | FID (20 steps) | FID (100 steps) |
|---------|----------------|-----------------|
| 0.1 | $8.93_{\pm 0.04}$ | $5.22_{\pm 0.02}$ |
| 0.2 | $7.81_{\pm 0.04}$ | $4.75_{\pm 0.02}$ |
| 0.3 | $\mathbf{7.48}_{\pm 0.05}$ | $4.53_{\pm 0.05}$ |
| 0.4 | $7.60_{\pm 0.05}$ | $4.54_{\pm 0.01}$ |
| 0.5 | $7.66_{\pm 0.03}$ | $4.49_{\pm 0.02}$ |
| 0.6 | $7.70_{\pm 0.05}$ | $4.47_{\pm 0.03}$ |
| 0.7 | $7.93_{\pm 0.05}$ | $4.59_{\pm 0.02}$ |
| 0.8 | $7.77_{\pm 0.05}$ | $\mathbf{4.42}_{\pm 0.02}$ |
| 0.9 | $8.09_{\pm 0.04}$ | $4.56_{\pm 0.02}$ |
| 1.0 | $8.35_{\pm 0.03}$ | $4.66_{\pm 0.04}$ |
| 1.1 | $8.60_{\pm 0.04}$ | $4.80_{\pm 0.03}$ |
| Baseline | $8.42_{\pm 0.07}$ | $4.63_{\pm 0.05}$ |

*Figure 17.* Complete FID scores on CIFAR-10 for all $\beta$ values. Our method reached the best validation FID after 320k steps, while the baseline took 340k. We used those checkpoints for the evaluation. We evaluated the FID using 5 seeds and report the mean as well as the standard deviation. Red denotes 20 step FID, blue 100 step FID, dotted line refers to baseline.

| | MNIST | | | | CIFAR-10 | | | |
|--------|-------|-------|----------------------|------------|-------|-------|----------------------|------------|
| Method | W1 $\downarrow$ | KS $\downarrow$ | Std ($\rightarrow$ 1) | Sinkhorn $\downarrow$ | W1 $\downarrow$ | KS $\downarrow$ | Std ($\rightarrow$ 1) | Sinkhorn $\downarrow$ |
| Gauss Noise | 0.922 | 0.651 | 26.37 | 705.5 | 0.397 | 0.179 | 2.02 | 1841 |
| Quant Noise | **0.310** | **0.522** | **5.93** | **114.7** | **0.311** | **0.159** | **1.83** | **1558** |

*Table 2.* Distribution-matching metrics comparing Gaussian and Quantile noise. Lower is better for W1, KS, and Sinkhorn; Std ratio should approach 1.

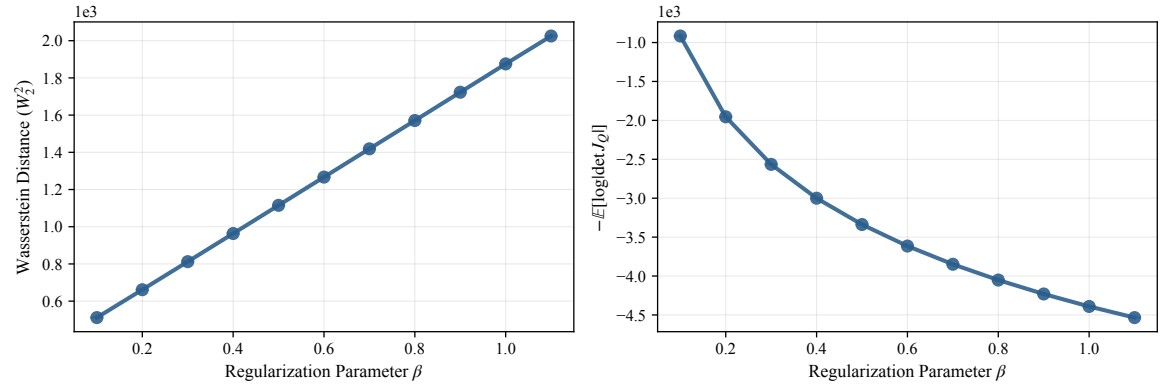

(f) Final performance metrics at 55k training steps vs. regularization parameter $\beta$.

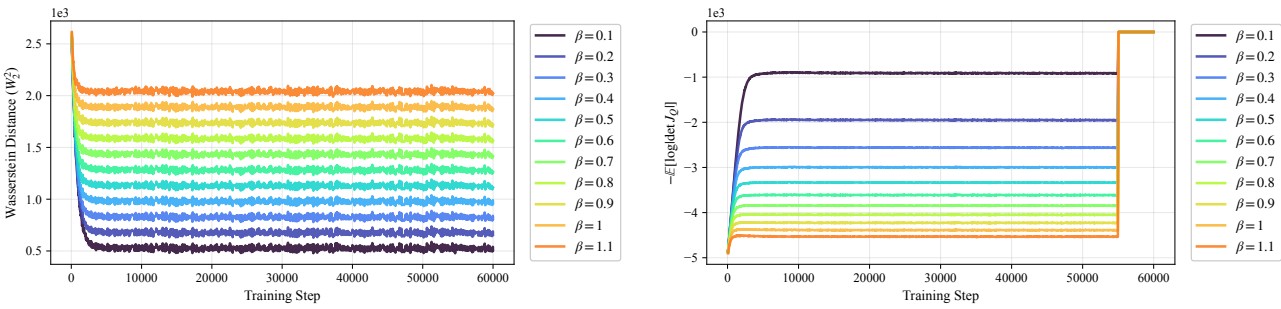

(b) Wasserstein distance evolution during training for different $\beta$.

(c) Regularization loss $-\mathbb{E}[\log | \det J_Q|]$ across $\beta$ values.

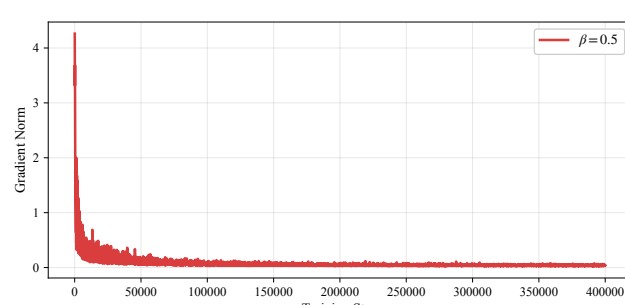

(d) Gradient norm of the quantile function during training.

(e) Gradient norm of the velocity field for fixed $\beta = 0.5$ over training.

*Figure 18.* Ablation studies showing the effect of regularization and model capacity on training dynamics and final performance.

