# OpenReview forum: "Adapting Noise to Data: Generative Flows from Learned 1D Processes"
_ICML.cc/2026/Conference — ICML 2026 regular_

### Official Review · Reviewer_qSep · 2026-03-07

**Soundness:** 3
**Presentation:** 3
**Significance:** 3
**Originality:** 3
**Overall Recommendation:** 5
**Confidence:** 4

**Summary:**

The authors propose to augment standard flow matching (FM) methods by optimizing over the source measure. They do this by optimizing over product source measures where $\mu = \prod_{i=1}^d \mu_i$  and propose to learn each component separately. This is achieved by expressing each component $\mu_i$ via a rational-quadratic-spline type parametrization which is easy to optimize over combined with a penalty term in the usual FM objective. They demonstrate the applicability of their algorithm on several synthetic and image-based generative modeling problems.

**Compliance With Llm Reviewing Policy:**

Affirmed.

**Key Questions For Authors:**

I only have a few minor questions and comments:
- Can you comment on the "stopgrad" aspect for jointly training these two neural networks? In particular, is there ever any hope that procedure will converge to a sufficiently nice optima?
- I think "narrow convergence" does not need to be mentioned in the background since this is never mentioned at any other point in the article
- The labeling in figure 5 could be a bit clearer.
- Many useful figures are deferred to the appendix. If possible, it would be good to move them into the main text.

**Limitations:**

yes

**Strengths And Weaknesses:**

The paper is well-written and proposes a mathematically principled approach to learning prior noise distributions for generative modeling. I believe the paper introduces some nice new mathematical objects and perspectives into the literature, and the experiments are sufficiently convincing.  The idea is also quite original, and has the potential for impact though unclear from the present experiments.

---

> ### Author Rebuttal · Authors · 2026-03-31
>
> We thank the reviewer for the positive evaluation and the helpful
> suggestions. Thank you for recognizing the mathematical originality and potential.
>
> > **"Can you comment on the 'stopgrad' aspect for jointly training these two
> neural networks? In particular, is there ever any hope that procedure will
> converge to a sufficiently nice optima?"**
>
> The stop-gradient on $(X_1 - X_0)$ means that the quantile network only receives gradients through the interpolated states $X_t = (1-t)X_0 + tX_1$. This discourages trivial collapse, since the quantile cannot reduce the FM loss simply by shrinking the endpoint displacement. In practice, this leads to a stable division of roles. The quantile learns a stable marginal transformation, while the velocity field learns the transport on top of it. We additionally freeze the quantile after a fraction of the training iterations. After which training reduces to standard FM with a fixed learned latent distribution. Empirically the quantile settles to a stable solution, as supported by the reported training stability and FID results.
>
> Thank you for the **helpful revision suggestions**. We will address all three points in a revised version. Furthermore we would like to note we have added a **100D funnel** experiment (see response to Reviewer mBFm) as well as the following real-world experiment.
>
>
> # Preliminary Heavy-Tailed Experiment - HRRR64
> We include a high-dimensional weather dataset with strongly heavy-tailed precipitation statistics. All models use the same UNet architecture (approximately 35M parameters). We evaluate on the total-precipitation channel of HRRR64-Mini (4×64×64 samples) [1] using 20,000 generated samples. This experiment is designed to test tail modeling in a realistic high-dimensional setting. While a heavy-tailed Student-t noise baseline consistently improves over the Gaussian baseline, the learned quantile noise performs best overall, yielding the strongest tail-frequency, tail-magnitude, and spectral results.
> | Model | Extreme-event frequency error ↓ | Extreme-event magnitude error ↓ | Spectral distance ↓ | Tail KS distance ↓ | Kurtosis deviation ↓ | Skewness deviation ↓ |
> |---|---:|---:|---:|---:|---:|---:|
> | **Gaussian baseline** | 0.9689 | 0.2455 | 3.1836 | 0.2067 | 4.93 | 1.157 |
> | **Student‑t baseline $\nu=4$**    | 0.8859 | 0.1482 | 2.0719 | 0.1014 | 2.89  | 0.830 |
> | **Quantile (Ours)** | **0.7550** | **0.0634** | **1.1063** | **0.0393** | **1.588** | **0.585** |
>
>
> Extreme-event frequency error: relative error in $P(X > q_{0.999})$ between generated and real data. Extreme-event magnitude error: relative error of the conditional mean above the $99.9$th percentile. Spectral distance: $L_1$ distance between radially averaged log power spectra (excluding DC). Tail KS distance: mean KS statistic restricted to beyond the $0.1$st/$99.9$th percentile. Kurtosis deviation: $|1 - \kappa_{\mathrm{gen}}/\kappa_{\mathrm{real}}|$. Skewness deviation: $|1 - \gamma_{\mathrm{gen}}/\gamma_{\mathrm{real}}|$. Lower is better for all metrics. Kurtosis deviation (KR), Skewness deviation (SR), and KS statistic as in [2].
>
>
> [1] NVIDIA Modulus, HRRR-Mini dataset: https://catalog.ngc.nvidia.com/orgs/nvidia/teams/modulus/resources/modulus_datasets-hrrr_mini
>
> [2] Pandey et al., Heavy-tailed diffusion models, 2024.

---

> > ### Author Rebuttal · Reviewer_qSep · 2026-04-03
> >
> > Thank you for addressing my few comments. I think my score is already quite high, and I will maintain it.

---

### Official Review · Reviewer_ng9u · 2026-03-12

**Soundness:** 3
**Presentation:** 3
**Significance:** 2
**Originality:** 2
**Overall Recommendation:** 4
**Confidence:** 4

**Summary:**

Isotropic Gaussian distributions are the standard latent distributions used in modern generative models. However, this prescriptive choice can be detrimental when trying to learn to generate from some distributions, e.g., heavy-tailed ones. In this paper, the authors propose to learn a data-adaptive noise distribution parametrized by one-dimensional quantile functions. Their method is computationally cheap, adaptable to any flow-based generative model, and leads to improved performance in unconditional generation tasks.

**Compliance With Llm Reviewing Policy:**

Affirmed.

**Final Justification:**

Authors addressed most of my concerns. Although I still see limitations/weaknesses in the paper I slightly lean towards an acceptance.

**Key Questions For Authors:**

1. What are the motivations for this method in high-dimensional settings? The limitations / shortcomings of using a Gaussian prior could be further expanded upon to make the contribution better motivated.

2. Could you present some more results in other high-dimensional or real-world settings? As this paper presents a methodological improvement, it is essential that the experiments on real data are either sufficiently diverse or sufficiently strong for the contribution to be significant.

3. How does one tune $\beta$ in practice?

If the authors address my concerns satisfactorily, I would be happy to improve my score.

**Limitations:**

Yes.

**Strengths And Weaknesses:**

**Strengths:**

- The paper is easy to follow.
- The method is simple and can be easily adapted to existing flow-based (unconditional) training algorithms.
- The contextualization and examples presented allow one to better understand how the parametrization via 1D quantile functions fits into the landscape of flow-based and score-based diffusion models.
- The numerical experiments on synthetic data are convincing.

**Weaknesses:**

Currently, the paper lacks sufficient evidence that the proposed method is practical.

- This approach effectively negates some of the nice properties of the latent space that one might rely on when using flow-based models beyond unconditional generation (e.g., for inpainting, fine-tuning, mixture of experts models). For instance, the support of the data-adaptive latent might not be 'convex', making it more difficult to move in the latent space.
- While the motivation is clear from the point of view of low-dimensional synthetic datasets (I find Neal's funnel to be a great illustration), it seems that the method has limited benefits in high-dimensional settings, due to the importance of the correlations between components. As the authors mention in the section on CIFAR-10 experiments, 'improvements are marginal as expected'. Combined with the previous point, this raises concerns about the effectiveness of the method.
- The high-dimensional experiments are few by modern ML standards and limited in scope.
- The $\beta$ parameter is not clearly introduced in the main body of the text. It should clearly appear in equation (16) along with the entropy regularization term. Moreover, there is no mention of how to choose its value a priori. For this method to be practical, there should be at least some heuristic for choosing $\beta$.
- (minor) Section 4.3 appears quite obscure to me and interrupts the flow of the paper (unless one is an expert in consistency models in particular). Since it mostly makes references to appendix C, I would suggest moving it completely to the appendix and mentioning it for readers who might be particularly interested in the connection between your work and DDIM.

---

> ### Author Rebuttal · Authors · 2026-03-31
>
> We thank the reviewer for the thoughtful and constructive feedback and appreciate the willingness to reconsider the score. We carefully address the raised concerns below.
>
> > **"This approach effectively negates some of the nice properties of the latent space… the support of the data-adaptive latent might not be 'convex'."**
>
> Each $Q_i$ is monotone by construction and continuous, so each marginal has interval support. Since the joint distribution is a product, its support is a product of intervals and hence convex.
>
> > **"It seems that the method has limited benefits in high-dimensional settings.."  /  "...motivations in high-dimensional settings?"**
>
> The benefit depends on how diverse the data marginals are across coordinates. On CIFAR-10 pixel marginals are very similar and the dominant structure is spatial correlation. There is not much to gain per coordinate and we are transparent about this. The gain is modest at 100 steps (FID 4.42 vs. 4.63) but more significant at 20 steps (7.48 vs. 8.42). We also tested a 55M parameter network ($\beta$ = 0.8) and observe 3.25 vs. 3.37 (100 steps) and 6.64 vs. 7.06 (20 steps).
>
> On data with diverse marginals the picture is different. We have added two new experiments. A 100D funnel experiment where coordinate 0 is Gaussian and coordinates 1–99 are heavy-tailed. Our method outperforms both Gaussian and Student-$t$ baselines (see response to Reviewer mBFm). Additionally we added a preliminary weather experiment (HRRR64, total precipitation) where variables have fundamentally different characteristics. Our method improves on all metrics including extreme-event frequency, tail distribution matching, and spectral distance (see response to Reviewer qSep).
>
> This paper establishes the idea and tools for noise design in FM. Fully exploiting this in very high dimensions is an open problem and we see this as important future work.
>
> > **"The β parameter is not clearly introduced… there is no mention of how to choose its value a priori." / "How does one tune β in practice?"**
>
>
> We agree that the presentation of $\beta$ should be improved and will make it appear explicitly in equation (16) in a revision. A comprehensive ablation across 11 values is provided in Figure 6 (full results in Figures 17/18).
>
> In practice, for all datasets, we sweep $\beta$ over a coarse grid ({0.1, …, 1.2}) during pretraining and choose the final value according to validation performance. No elaborate tuning is needed. $\beta$ controls the tradeoff between marginal adaptation and latent smoothness. On data with diverse marginals (e.g. MNIST) smaller $\beta$ lets the quantile adapt more aggressively. On data with similar marginals (CIFAR-10) larger $\beta$ keeps the latent smoother.
>
> The method is not overly sensitive. On CIFAR-10, 9 of 11 tested values outperform the Gaussian baseline (Figure 17), with $\beta$ = 0.3 best at 20 steps (FID 7.48, still competitive at 100 steps with 4.53) and $\beta$ = 0.8 best at 100 steps (FID 4.42).
>
> > **"Section 4.3 appears quite obscure… I would suggest moving it completely to the appendix."**
>
> Section 4.3 establishes that our quantile interpolants generalize the DDIM interpolants and satisfy the consistency properties needed for few-step models. We consider this relevant given the growing interest in few-step generation. We can shorten the main text and defer details to the appendix.

---

> > ### Author Rebuttal · Reviewer_ng9u · 2026-03-31
> >
> > I thank the authors for their response. I am glad to see additional experiments on weather data, and I am reassured that $\beta$ is not overly sensitive. I will increase my score.
> > I still have concerns about altering the latent space.
> > I admit that my concern about convexity was poorly phrased, as your method does induce a convex prior distribution. Essentially, I am curious to see what happens when one tries to combine different such models, how would one deal with the resulting prior distribution?

---

> > > ### Author Response · Authors · 2026-04-01
> > >
> > > Thank you for the follow-up and for increasing the score. For downstream applications (inpainting, fine-tuning, inverse problems, likelihood evaluation) our learned quantiles provide the same tools as the Gaussian case. Each $Q_i$ is invertible and the prior density is available analytically.
> > >
> > > For combining models with different learned priors (e.g. model merging) one cannot naively average velocities since the probability paths differ. However, if one were to interpolate linearly in U-space ($U_t = (1-t)U_0 + tU_1$, $X_t = Q(U_t)$), the velocity decomposes as
> > >
> > > $$\dot{X}_ t^i = Q_i'(U_t^i) \cdot (U_1^i - U_0^i),$$
> > >
> > > separating the U-space displacement $(U_1^i - U_0^i)$ from the geometry factor $Q_i'$. One could in principle train the network to predict the displacement in U-space rather than the data-space velocity. To translate between flows, one maps the state via $X_t^B = Q_B(Q_A^{-1}(X_t^A))$ and the velocity via
> > >
> > > $$v_B^i = \frac{Q_B^{i\prime}(U_t^i)}{Q_A^{i\prime}(U_t^i)} \cdot v_A^i,$$
> > >
> > > a coordinatewise rescaling by the ratio of quantile derivatives at the shared U-space point $U_t = Q_A^{-1}(X_t^A)$. All required operations (inverse, derivative) are available analytically through the RQS parameterization. Each quantile then induces a different Riemannian metric on data space ($g_{ii} = 1/Q_i'^2$) and this ratio is the metric correction. This is one way one could approach model combination with different learned priors. We find it interesting to think further in this direction but it is beyond the scope of this paper.
> > >
> > > If this is not what the reviewer had in mind we would be happy to discuss further.

---

### Official Review · Reviewer_mBFm · 2026-03-13

**Soundness:** 3
**Presentation:** 2
**Significance:** 2
**Originality:** 2
**Overall Recommendation:** 3
**Confidence:** 3

**Summary:**

The paper introduces a framework for learning the initial noise for flow-based generative models based on the data. The paper uses Rational Quadratic Splines to learn interpolations of the data to parametrise one-dimensional quantile functions, for each dimension of the noise. These are then combined to form the final multidimensional noise distribution under the independence assumption. The learning is done by minimising a Wasserstein distance between the noise and data. This reduces the length of the path the flow model needs to learn.

The method is evaluated on synthetic datasets, MNIST, and CIFAR-10, demonstrating improved performance in some scenarios (especially under limited model capacity) and minimal computational overhead.

**Compliance With Llm Reviewing Policy:**

Affirmed.

**Final Justification:**

The authors have provided satisfactory responses to the questions. The remaining concern pertains to the evaluation on small and low-resolution image datasets, which raises questions on the scalability of the method to higher dimensions. This is a weakness of the paper, particularly given that almost all of the state-of-the-art evaluations are carried out on high-resolution image datasets.

**Key Questions For Authors:**

1/ What is the justification for the choice of interpolation for the quantiles? Would other interpolations work?

2/ How much is the improvement of the method compared to baseline when allowed to train to saturation (using early stopping)?

3/ How does the method compare to other ways of learning the initial noise or noise distributions other than the Gaussian?

4/ What is the performance improvement of the method for larger networks and datasets?

**Limitations:**

There is no statement on limitations or negative societal impact. It is advisable to include such a section.

**Strengths And Weaknesses:**

**Strengths**

1. Clear motivation and well identified problem:

The paper motivates the problem well and presents convincing arguments for why Gaussian noise does not match well with most data distributions.

2. Lightweight Parametrisation with minimal overhead:

Rational Quadratic Splines provide an efficient way to learn the noise with a minimal number of additional parameters (300k for CIFAR). The empirical computational overhead is also reported to be small.

3. Good theoretical grounding

The paper provides a good theoretical grounding for their method. The paper is mathematically rigorous.





**Weaknesses**

1. Lack of proper experimental comparisons:

The paper lacks proper experimentation to compare its method against noise distributions other than the baseline (Gaussian). Some examples of distributions used as initial noise by literature (as mentioned in the introduction of the paper) are “Student's-t noise distribution with a tunable degrees-of-freedom parameter ν” and “α-stable noise”. However experimental comparisons are never made with these methods. This makes it difficult to evaluate the improvements provided by the method. Even the comparisons with baseline, in the main paper, are only presented through a single graph.

2. Poor performance of larger networks:

The performance improvement reduces as the network is made larger. This severely limits the practical applicability of the method, since most practically useful flow-generative models use large networks. More experiments are also needed on larger networks to check the practical usability of the method.

3. Independence assumption for the noise latent:

Images have high local pixel correlation. Thus, learning independent noise components would only allow one to learn the pixel-wise marginal distribution. This will provide limited benefit when trying to flow from noise to image data.

4. Joint training of Flow Model and Noise distribution:

Jointly training the starting noise and the vector field to follow given the staring points (as flow models do) could create a “moving the goal post” effect where the model needs the constantly optimise the flow for the new starting point learnt. Since learning the initial noise and the final flow model can be done independently, would suggest the authors to first learn the noise and then freeze the splines and lean the flow model.

5. Choice of interpolation:

Some justification for the choice of Rational Quadratic Splines (RQS) would strengthen the paper. Additionally, experimental comparisons against other methods of learning the initial noise (other interpolations or other methods from literature) are needed to justify the choice of RQS.

6. Modest conceptual contribution:

The way the noise is learnt is theoretically sound and lightweight but this is a small contribution, and the performance improvements are marginal.

7. Excessively large background section:

The background section goes into extreme detail regarding flow-matching and quantiles (Sections 3 and 4). Reducing this would leave more space for further experimental studies that the paper lacks.


Overall, the key contribution is that replacing fixed Gaussian noise with a learned quantile-based latent distribution can shorten transport paths and improve training efficiency in flow-based generative models while introducing minimal overhead. The idea is elegant and technically sound, but the empirical gains on realistic datasets are modest, and the independence assumption may limit broader applicability.

---

> ### Author Rebuttal · Authors · 2026-03-31
>
> We thank the reviewer for the feedback and for recognizing that "the idea is elegant and technically sound." We carefully address your concerns below.
>
> > **"The paper lacks adequate baseline comparisons…"**
>
> We have conducted two new experiment. On a **100d Neal funnel** and a second preliminary on  **HRRR64, total precipitation**, see our response to Reviewer qSep.
>
> ## 100D Neal Funnel
> We compare against Gaussian, Student-t ($\nu$=4), and (inspired from our 2d experiment) a Student-t ($\nu$=20 for dim 0, $\nu$=4 for dims 1+). All methods use MLP (256×3)  trained for 500k steps. We swept $\beta$ over {0.1, …, 1.2}.
>
> | Method | MMD ↓ | KS (mean) ↓ | Tail KS (mean) ↓ | Tail KS (max) ↓ |
> |--------|-------|-------------|-------------------|-----------------|
> | Ground truth | 0.0001 | 0.0012 | 0.0406 | 0.0634 |
> | Gaussian | 0.0105 | 0.0786 | 0.1628 | 0.2051 |
> | Student-t (ν=4) | 0.0181 | 0.0825 | **0.0791** | 0.1403 |
> | Student-t (ν=20/4) | 0.0174 | 0.0806 | 0.0815 | 0.1444 |
> | **Ours (β=0.5)** | **0.0073** | **0.0739** | 0.0971 | 0.1650 |
> | **Ours (β=0.7)** | 0.0088 | 0.0834 | 0.1081 | **0.1386** |
>
> MMD: energy distance (50k samples). KS: mean two-sample Kolmogorov–Smirnov statistic (1M samples). Tail KS: KS restricted to beyond the 0.1st/99.9th percentile. Ground truth shows the noise floor.
>
> > **"How does the method compare to other ways of learning the initial noise.."**
>
> Existing methods for learning the noise (e.g. [1], [2]) involve substantially more complex training. A direct comparison is not straightforward.
>
> > **"What is the performance improvement of the method for larger networks and datasets?"**
>
> We have additionally tested a larger 55M parameter network. With $\beta$ = 0.8 we observe FID 3.25 vs. 3.37 (Gaussian baseline) at 100 steps and 6.64 vs. 7.06 at 20 steps on CIFAR10.
>
> > **"How much is the improvement compared to baseline when allowed to train to saturation..."**
>
> We evaluate at the best validation FID checkpoint for both methods (ours at 320k, baseline at 340k, 5 seeds). The numbers in Figure 6 (4.42 vs. 4.63) reflect saturated performance.
>
> > **"... learning independent noise components would only allow one to learn the pixel-wise marginal distribution."**
>
> This is not quite accurate. Because we optimize via global OT coupling (not per-coordinate), the learned quantile is in general not the product of data marginals. We give an explicit example in the paper (Example E.1) where the $W_2$ minimal product measure differs from the 1d marginals. The OT coupling introduces cross-coordinate information. That said, the learned noise cannot represent correlations directly. The velocity field handles those.
>
> > **"Jointly training the starting noise and the vector field could create a 'moving the goal post' effect…"**
>
> We tested both joint training and pretraining followed by freezing. Both work well. The stop-gradient on ($X_1$ − $X_0$) means the quantile only receives gradients through $X_t$, and we freeze the quantile early in training. The gradient through $X_t$ is somewhat heuristic but training is empirically stable (Appendix, Figure 18).
>
> > **"What is the justification for the choice of interpolation for the quantiles? ... "**
>
> In 1D the quantile function is the OT map from $\mathcal{U}(0,1)$ to the target. RQS [3] are universal approximators for monotone functions, exactly invertible, cheap to evaluate, and admit analytic log-derivatives for the entropy regularization. Any monotone parameterization would work. Our framework is agnostic to this.
>
> > **"The way the noise is learnt is theoretically sound and lightweight but this is a small contribution."**
>
> The main point is establishing noise design as a tractable degree of freedom in FM via the 1D quantile decomposition. The 1D viewpoint makes noising processes accessible that have no straightforward multi-dimensional analogue (Kac, MMD gradient flow). These illustrate the richness of the design space. The quantile parameterization sidesteps the choice of process by learning the noise from data, important when Gaussian noise is inadequate (e.g. heavy-tailed targets, see also [4], [5]).
>
> > **"The background section goes into extreme detail…"**
>
> The detail is deliberate. It makes the development self-contained, and processes like Kac and MMD gradient flow are not widely known. We can move some material to the appendix in a revision.
>
> > **"There is no statement on limitations or negative societal impact."**
>
> This was an oversight and will be added in a revision. We do not see specific societal consequences that must be highlighted.
>
> [1] Bartosh et al., Neural flow diffusion models: Learnable forward process for improved diffusion modelling, 2024.
>
> [2] Sahoo et al., Diffusion models with learned adaptive noise, 2023.
>
> [3] Durkan et al., Neural spline flows, 2019.
>
> [4] Pandey et al., Heavy-tailed diffusion models, 2024.
>
> [5] Tam & Dunson, On the statistical capacity of deep generative models, 2025.

---

> > ### Author Rebuttal · Reviewer_mBFm · 2026-04-05
> >
> > I appreciate the response provided by the authors. The additional experimental results strengthen the paper.
> >
> > The image results reported are on low-resolution datasets (MNIST and CIFAR10). CelebA would have been a better choice to demonstrate that the proposed method does not have any issues scaling to high dimensions. Barring this concern, the authors have provided satisfactory answers to the other questions. I will increase my score.

---

### Official Review · Reviewer_z7FN · 2026-03-17

**Soundness:** 3
**Presentation:** 3
**Significance:** 3
**Originality:** 3
**Overall Recommendation:** 5
**Confidence:** 3

**Summary:**

This paper argues that generative flow models shouldn’t start from a fixed Gaussian noise distribution but instead learn a noise distribution tailored to the data. They do this by parameterising noise via a simple 1D quantile functions and building high-dimensional noise from these components. By making the noise already resemble the data, the model only needs to learn a much simpler transformation. This leads to shorter transport paths, and better performance in low-step generation settings. The trade-off is that their approach simplifies dependencies by treating dimensions mostly independently.

**Compliance With Llm Reviewing Policy:**

Affirmed.

**Ethical Review Concerns:**

N.A

**Key Questions For Authors:**

See weaknesses.

**Limitations:**

Yes.

**Strengths And Weaknesses:**

Strengths:

- Elegant reduction to 1d quantile functions. They are monotonic and universal in 1D, giving a stable parameterisation.
- By decomposing into 1d components problems, training is quite light weight so they achieve computational efficiency and scalability.
- This framework of learning the noise from data fits naturally into FM pipelines.
- Clearly presented.

Weaknesses:

- The obvious one is the independence assumption of the 1d quantile function construction, the distortions of marginals.
- It would be good to get a sense of the error introduced by the independence assumption.
- The experiments compare learned noise with fixed Gaussian noise which is a decent apples-to-apples comparison but what about other stronger baseline approaches like DDPM++ and other hybrid flow diffusion models.

---

> ### Author Rebuttal · Authors · 2026-03-31
>
> We thank the reviewer for the positive assessment and recognition of our framework. We address the remaining points below.
>
> > **"The obvious one is the independence assumption of the 1d quantile function construction, the distortions of marginals."**
>
> We agree that the independence assumption is the central structural limitation. We would like to point out that the standard Gaussian $\mathcal{N}(0, I_d)$ is also a product measure. Our method adapts each marginal to the data, capturing per-coordinate scales, tail behavior, and support constraints. The velocity field handles cross-dimensional structure, as in standard FM. The MNIST capacity ablation (Figure 4) shows this division of labor in action. At reduced capacity, our learned latent maintains performance while the Gaussian baseline degrades. This indicates that offloading marginal structure to the noise frees up network capacity for correlations.
>
> > **"It would be good to get a sense of the error introduced by the independence assumption."**
>
> We want to clarify that the product measure is the noise distribution, not an approximation of the data, so there is no "error" in the usual sense. All cross-coordinate structure is handled by the velocity field. The relevant question is how much the learned product noise helps compared to the Gaussian product noise. Which is directly measured by FID, large gains on data with strong marginal structure, modest gains where correlations dominate.
>
> To quantify the distributional fit of the learned latent directly, we compare against a standard Gaussian on both per-dimension marginal statistics and a global multivariate metric. We report the 1D Wasserstein distance (W1) and Kolmogorov–Smirnov statistic (KS), both averaged over dimensions, the ratio of generated to data standard deviation per dimension (Std ratio, ideal value 1), and the Sinkhorn divergence computed on full latent vectors (measuring global distributional mismatch). The quantiles were trained with $\beta = 0.1$ on MNIST and $\beta = 0.8$ on CIFAR-10:
>
> | Dataset | Metric | Quantile | Gaussian |
> |---------|--------|----------|----------|
> | MNIST | W1 ↓ | 0.310 | 0.922 |
> | MNIST | KS stat ↓ | 0.522 | 0.651 |
> | MNIST | Std ratio (→1) | 5.93 | 26.37 |
> | MNIST | Sinkhorn ↓ | 114.7 | 705.5 |
> | CIFAR-10 | W1 ↓ | 0.311 | 0.397 |
> | CIFAR-10 | KS stat ↓ | 0.159 | 0.179 |
> | CIFAR-10 | Std ratio (→1) | 1.83 | 2.02 |
> | CIFAR-10 | Sinkhorn ↓ | 1558 | 1841 |
>
>
> > **"What about other stronger baseline approaches like DDPM++ and other hybrid flow diffusion models."**
>
> Our contribution is a modular replacement for the latent distribution, not a new architecture. It is designed to plug into any existing FM pipeline without changing the backbone, so the natural comparison is Gaussian vs. learned latent in the same setup. On CIFAR-10 with the U-Net from [1] / [2] (~35M params, chosen to match the reference values in [1]) we observe FID 4.42 vs. 4.63. To verify that gains persist with stronger backbones, we tested a larger 55M parameter network. Our method (with $\beta = 0.8$) achieves FID 3.25 vs. 3.37 at 100 steps, and 6.64 vs. 7.06 at 20 steps (compared to Gaussian baseline).
>
> We would also like to note that we have conducted a new **100D funnel** experiment (see response to Reviewer mBFm) and a preliminary experiment on weather data (**HRRR64**, total precipitation, see response to Reviewer qSep).
>
> [1] Tong et al., Improving and generalizing flow-based generative models with minibatch optimal transport, 2023.
>
> [2] Nichol & Dhariwal, Improved denoising diffusion probabilistic models, 2021.

---

> > ### Author Rebuttal · Reviewer_z7FN · 2026-04-04
> >
> > Thank you for the response, I better understand the contribution of the paper in light of the performance with stronger architectural backbones. I would maintain my score.

---

### Decision · Program_Chairs · 2026-04-30

**Decision:**

Accept (regular)

**Comment:**

The main concern of this paper is the experiments, which were carried out on relatively low-dimensional data. However, reviewers agreed this was an elegant solution to a well-motivated problem, which outweighed the concerns.

An acknowledgment of possible scaling issues to high-dimensional data in the manuscript would be appropriate, and the inclusion of results reported during rebuttal.